# Smoke-free status of homes and workplaces among Indian people: Evidence from Global Adult Tobacco SurveyData-2016/2017

**Mohammad Ali**[1◎], **Most. Farida Khatun**[2◎], **Tasnuva Yasmin**[3◎], **Ashis Talukder**[1◎], **Md. Maniruzzaman**[1◎], **Sharlene Alauddin**[1◎]*

**1** Statistics Discipline, Khulna University, Khulna, Bangladesh, **2** Pharmacy Discipline, Khulna University, Khulna, Bangladesh, **3** Development Studies Discipline, Khulna University, Khulna, Bangladesh

◎ These authors contributed equally to this work.
* sharlene.kst@gmail.com

**Data Availability Statement:** The data set used in this study is publicly available from the website of Centers for Disease Control and Prevention

## Abstract

### Aims

This study aimed to determine the impact of correlates on tobacco control/smoke-free status of homes and workplace among Indian people. To assess the magnitude of the problem, the relationship between smoke-free status and secondhand smoke (SHS) exposure was also explored.

### Methods

Data was extracted from the Global Adult Tobacco Survey Data (GATS)-2017. It was a household survey that included people aged 15 years or older and covered all 30 states and 2 Union Territories (UTs) of India. A logistic regression model was used to determine the correlates of smoke-free status of homes and workplaces. Additionally, the Pearson correlation was used to explore the relationship between smoke-free status and the proportion of participants exposed to SHS both at homes and in the workplaces.

### Results

The overall prevalence of smoke-free status in the home and workplace was 62.8% and 51.7%, respectively. Results of multivariate analysis (Logistic regression) illustrated that indicators like tobacco smoking status, place of residence, region, education, occupation, wealth quintile, and knowledge status about children's illness were significantly associated with the respondent's intention to live in a completely smoke-free environment both at home and in the workplace in India. This study revealed that SHS exposure was significantly negatively associated with a smoke-free status.

### Conclusion

This study will help the policymakers to promote efficient policies for improving smoke-free status and to ensure a better environment both at home and in the workplace in India.

(CDCP). In order to gain access to the data files, researchers should complete the form here (https://nccd.cdc.gov/GTSSDataSurveyResources/Ancillary/DataReports.aspx?CAID=2).

**Funding:** The authors received no specific funding for this work.

**Competing interests:** The authors have declared that no competing interests exist.

# 1 Introduction

Tobacco use is a major risk factor for non-communicable diseases and one of the top preventable causes of worldwide premature deaths [1]. All forms of tobacco products, whether they are smoked, inhaled, chewed or sucked, cause serious damage to health [2]. Tobacco kills more than 8 million people each year globally, and among them, 7 million people die due to direct tobacco use, while around 1.2 million deaths occur due to secondhand smoke (SHS) [3]. India is the second-largest consumer of tobacco after China in the world and one-third of its population uses tobacco [4,5]. The reports of the Global Adult Tobacco Survey 2 (GATS2)-2017 stated that 266.8 (28.6%) million adults use tobacco in any form in India, whereas 10.7% of people smoked tobacco and 21.4% of people use smokeless tobacco [6]. Additionally, the prevalence of current tobacco usage among males is high (42.4%), compared to females (14.2%) [6]. According to the GATS 2 (2016/2017) report, a total of SHS, 38.7% (38.1% male vs. 39.3% female) of people do smoking at home, 30.2% (32.7%male vs.17.9% female) at the workplace and 5.3% (8.1% male vs. 2.4% female) at the government office, 5.6% (6.8% male vs. 4.4% female) at health care facilities, 7.4% (13.0% male vs. 1.6% female) at restaurants [6]. Asian women have a lower tendency to smoke while most of them and their children are mainly victims of SHS and suffer from various smoking-induced harms [7]. According to previous studies, about one-third of non-smokers are adults and 40% of children are exposed to SHS [7,8]. Several studies showed that SHS increased the risks of lung cancer, stroke, bronchitis, childhood diseases, and heart disease [9–11].

For the strong implementation of smoking policies and to protect non-smokers from SHS, legislation was enacted by the parliament of India in 2003 entitled "Cigarettes and Other Tobacco Products Act 2003" (COTPA 2003) which prohibits smoking in public and work premises [12]. The provisions under this law were notified to the public from 2004 to 2006. The Government faced many legal challenges from the tobacco industry during this period. However, in 2008, after a long legal fight and intervention of civil society, a remarkable effect of these rules was found both in public and workplaces [38]. Additionally, to fulfill the commitments under the WHO Framework Convention on Tobacco Control (FCTC) and to facilitate the effective implementation of COTPA 2003, the Ministry of Health and Family Welfare, Government of India, launched the National Tobacco Control Program in 2007–2008 [12]. Although there is progress in smoke-free status both in indoor public/workplaces and public transportation, the policies are still not comprehensive [6,13]. All of these rules come to the public as a Government or administrative order and there is a lack of awareness of following any legal instruments. Moreover, the law is not so extensive as it permits smoking facilities with designated smoking areas in places like offices, large restaurants and hotels where the penalty for such violations is modest (200 rupees fine–US $3.80) [20]. However, potential failure in implementing smoke-free legislation in public/workplaces could cause a compensatory increase in smoking at home and increase SHS exposures in the family. Tobacco control efforts encourage people to decrease overall smoking at home and the workplace. Without strong enforcement and clear guidelines for the citizens about their right to smoke, the implementation of these policies will remain to be largely ineffective [39]. It is essential to speed up the achievements and target of a successful implication of a smoke-free policy in India.

Several studies have been conducted to reflect tobacco smoking status in India [14–17]. However, most of these previous studies have only focused on the prevalence and predictors of tobacco use [14–17]. Another study investigated SHS exposure at home and in public places among smokers and non-smokers [18]. A study also observed that a smoke-free workplace was significantly associated with SHS exposure at home [19,20]. Previous research showed that the presence of policies that restrict smoking in the workplace and home may be varied by

geographical region, occupation, and respondent's socioeconomic status [21]. Noticeably, no previous study focused on the risk factors affecting the functionality of smoke-free status both at home and workplace in India. To our knowledge, this is the first study in India to determine the influential correlates of smoke-free status both at home and in the workplace respectively. The proper implications of a smoke-free policy may work as an effective intervention in reducing the practice of smoking. Additionally, this study explores the relationship between smoke-free status and SHS exposure by measuring the correlation between the state-level proportion of respondents at both premises. This association is only measured to investigate the link of SHS exposure to smoke-free law coverage, which aims to protect SHS from the danger of passive smoking.

## 2 Materials and methods

### 2.1 Data

The data was extracted from the Global Adult Tobacco Survey 2 (GATS 2), which was conducted in India between August 2016 and February 2017. A detailed description of the survey objectives, design, methods, and questionnaires can be found elsewhere [6]. In brief, GATS2 is a global standard for systematically monitoring the status of using tobacco among adults and tracking key control policies. GATS 2 conducted a household survey of people aged 15 years or older, covering all 30 states and 2 Union Territories (UTs) in India. A multistage stratified cluster sampling design was applied to collect the sample from each state as well as particularly from urban and rural areas. In GATS 2, a total of **74,037** individual interviews were completed with an overall response rate of 92.9%. The analysis of the smoke-free status of the workplace was restricted to the GATS2 respondents who were working indoors or both indoors and outdoors. After removing the respondents with missing values, the final analysis of the smoke-free status was conducted with **71,046** respondents from home and **15,254** from workplaces. The flowchart of sample selection procedures for both smoke-free status of homes and workplaces is shown in Fig 1.

### 2.2 Ethics approval and consent to participate

This study utilized a secondary dataset that is publicly available. The GATS2-2016/2017 survey was approved by the Ministry of Health and Family Welfare, Government of India. The whole data was completely anonymized, de-identified, and aggregated before access and analysis.

### 2.3 Study or dependent variable

In this work, we considered smoke-free practices both at home and the workplace as our main study or dependent variable. It is classified into binary categories in the following ways: the respondent was classified as having a complete smoke-free policy at home if he/she answered "smoking is never allowed inside your home" to the question "Which of the following best describes the rules about smoking inside your home?" If the answer was "smoking is allowed inside your home," "smoking is generally not allowed but exceptions," or "there are no rules about smoking at your home" he/she was considered as not having a complete smoke-free policy at home.

Similarly, the respondent was classified as working in a completely smoke-free environment if he/she answered "not allowed in any indoor areas" to the question "Which of the following best describes the indoor smoking policy where you work?" If the answer was "smoking is allowed anywhere", "allowed only in some indoor areas", or "there is no policy", he/she was considered as not working in a completely smoke-free environment.

**2.3.1 Secondhand Smoker (SHS).** In this study, the term 'Secondhand smoker (SHS)' was only considered to measure the state-level correlation between the proportion of smoke-free

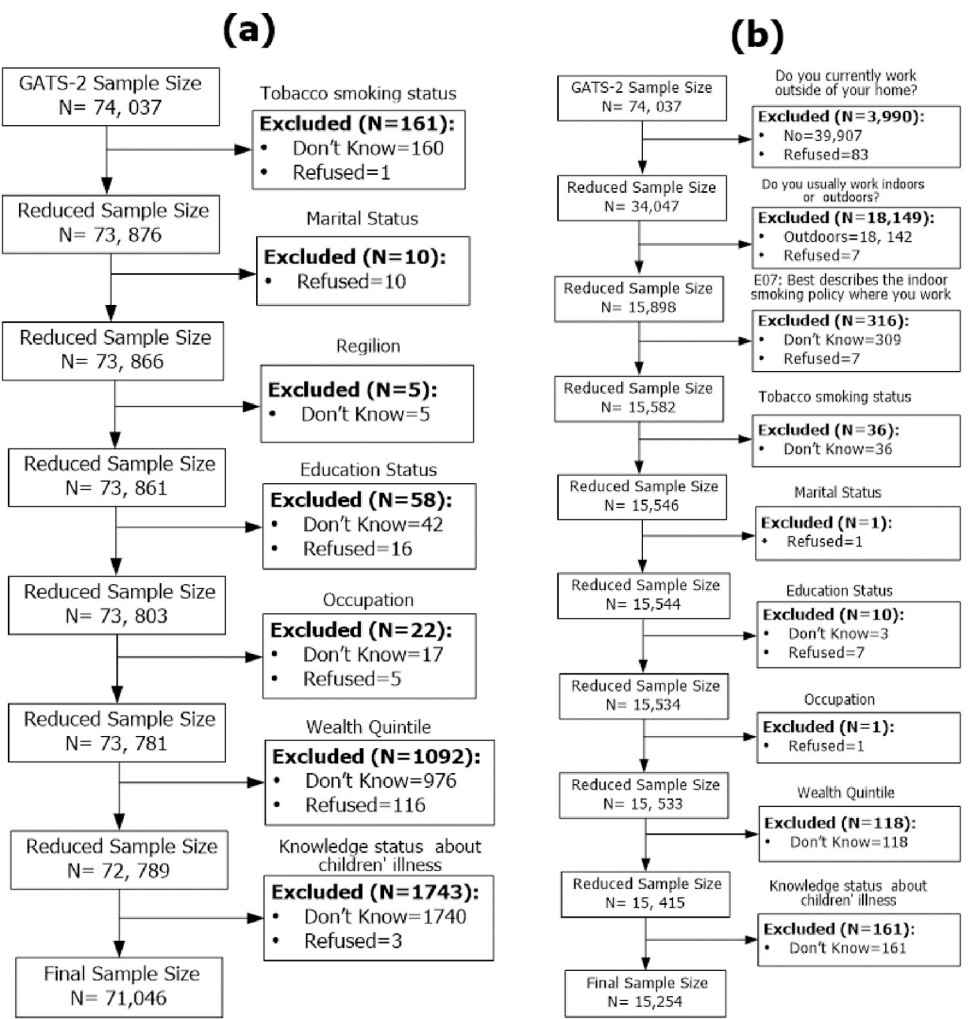

**Fig 1.** Flowchart of sample selection for smoke free status of: (a) Home and (b) Workplace.

status and SHS exposure. In GATS 2, to measure the SHS exposure both at home and workplace the following questions were considered: (i) 'How often does anyone smoke inside your home? Would you say daily, weekly, monthly, less than monthly, or never?' Respondents who indicated 'daily', 'weekly', or 'monthly' were classified as exposed to SHS in the home; (ii) For the question, during the past 30 days, did anyone smoke in indoor areas where you work?', respondents who answered 'yes' were classified as exposed to SHS in the workplace.

## 2.4 Variables of the study

In this study, to construct the wealth quintile variable, the following indicators were used: Whether the household or person living in the household enjoys the facilities of i) electricity, ii) electric fan, iii) air conditioner, iv) refrigerator, v) washing machine, vi) radio, vii) television, viii) computer/laptop, ix) internet connection, x) moped/scooter xi) car, xii) flush toilet, xiii) fixed telephone, xiv) cell telephone. Principal component analysis (PCA) was conducted to create this variable and it was divided into five categories (poorest, poorer, medium, richer, and richest) based on the quintiles of the first PCA scores. The estimated proportion of participants living in smoke-free homes and workplaces was collected and the state-level correlation

**Table 1. Descriptions of the independent variables along with their categorizations.**

| Factors | GATS asked the following questions | Categorizations |
|---|---|---|
| Tobacco smoking status | Does this person currently smoke tobacco, including bidis, cigarettes, hukkah, cigars, and pipes? | 1 = Smoker, 2 = Non-smoker |
| Age (in years) | What is the age of the selected person? | 1 = 15–30,2 = 31–45,3 = 46–60,4 = More than 60 |
| Gender | Record the gender of the selected person | 1 = Male, 2 = Female |
| Marital status | What is your marital status? | 1 = Single, 2 = Married, 3 = Separated/others |
| Religion | What is your religion? | 1 = Hindu,2 = Muslim,3 = Christian,4 = Others |
| Placeofresidence | Residence status | 1 = Urban,2 = Rural |
| Region | National geographical regions | 1 = North, 2 = Central, 3 = East, 4 = North-east, 5 = West, 6 = South |
| Education | What is the highest level of education you completed? | 1 = No education,2 = Up to primary, 3 = Up to secondary, 4 = More than secondary |
| Occupation | Which of the following best describes your main work status over the past 12 months? | 1 = Govt./Non-Govt.employee,2 = Dailywise/causallaborer,3 = Self-employed,4 = Homemaker, 5 = Student, 6 = Others |
| Wealth quintile | How would you describe the status of your Wealth quintile? | 1 = Poorest, 2 = Poorer, 3 = Middle, 4 = Richer, 5 = Richest |
| Knowledgestatus aboutchildren' illness | Based on what you know or believe, does smoking tobacco cause serious illness among the children? | 1 = No, 2 = Yes |

between the proportion of smoke-free status and SHS exposure at both home and workplace was also explored. Reviewing some pre-existing works of literature, other relevant explanatory variables (given in Table 1) were selected [5,9,7,15,17,18]. The descriptive summary of the independent variables along with their categorizations is presented in Table 1. In addition, the estimated proportion of participants living in smoke-free homes and workplaces was used for exploring the state-level correlation between the proportion of smoke-free status and SHS exposure both at home and in the workplace.

## 2.5 Statistical analysis

A descriptive analysis was performed and manifested the frequency (percentage) of smoke-free status both at home and workplace. A Chi-square test was conducted to assess the bivariate association between the dependent and independent variables. For multivariate analysis, a logistic regression model was used to estimate the adjusted odds ratio (AOR) of the selected independent variables along with their correspondence p-values and 95% confidence interval. We determined the most significant variables of smoke-free status both at home and workplace using a p-value ($<0.001$). In this study, the multicollinearity among different independent variables was checked by variance inflation factor (VIF) values. The VIF values less than 10 indicated, no existence of significant multicollinearity between the independent variables [22]. The entire data analysis, data cleaning, and management were carried out using STATA version 12 (StataCorp LP, College Station, TX, USA). Sampling weights were applied and the weighted estimates were calculated to consider the complex study design. Clustering and stratification were also counted with the svyset command in STATA. The following variables were used to apply clustering and stratification: gatscluster, gatsstrata and gatsweight. A Pearson correlation test was carried out to explore the relationship between smoke-free status and the proportion of SHS exposure at both home and workplace.

## 3 Results

### 3.1 Prevalence of complete smoke-free practices in home and workplace

The overall prevalence of complete smoke-free status at home and workplace were 62.8% and 51.7% (See Table 2). About 25% and 48.5% of tobacco smokers follow smoke-free status in

**Table 2. Correlates of smoke-free status of homes and workplaces in India.**

| Characteristics | Smoke-free status in | | | | | |
|---|---|---|---|---|---|---|
| | Home (n = 15,254) | | | Workplace (n = 71,046) | | |
| | Yes % | No, % | p-value | Yes, % | No, % | p-value[1] |
| Overall | 62.8 | 37.2 | | 51.7 | 48.3 | |
| **Tobacco smoking status** | | | <0.001 | | | <0.001 |
| Smoker | 28.5 | 71.5 | | 48.5 | 51.5 | |
| Non-smoker | 56.1 | 43.9 | | 66.9 | 33.1 | |
| **Age (in year)** | | | 0.944 | | | 0.676 |
| 15–30 | 51.8 | 48.2 | | 63.6 | 36.4 | |
| 31–45 | 51.8 | 48.2 | | 61.8 | 38.2 | |
| 46–60 | 51.3 | 48.7 | | 62.7 | 37.3 | |
| More than 60 | 51.5 | 48.5 | | 63.2 | 36.8 | |
| **Gender** | | | 0.205 | | | <0.001 |
| Male | 52.2 | 47.8 | | 61.6 | 38.4 | |
| Female | 51.1 | 48.9 | | 68.9 | 31.1 | |
| **Religion** | | | <0.001 | | | 0.0086 |
| Hindhu | 51.9 | 48.1 | | 63.5 | 36.5 | |
| Muslim | 45.4 | 54.6 | | 57.1 | 42.9 | |
| Christian | 65.4 | 34.6 | | 64.6 | 35.4 | |
| Others | 35.0 | 65.0 | | 68.5 | 31.5 | |
| **Marital status** | | | <0.001 | | | <0.001 |
| Single | 56.0 | 44.0 | | 69.0 | 31.0 | |
| Married | 50.2 | 49.8 | | 60.9 | 39.1 | |
| Separated/others | 52.6 | 47.4 | | 58.9 | 41.1 | |
| **Place of residence** | | | <0.001 | | | <0.001 |
| Urban | 46.1 | 53.9 | | 56.0 | 44.0 | |
| Rural | 62.1 | 37.9 | | 70.8 | 29.2 | |
| **Region** | | | <0.001 | | | <0.001 |
| North | 56.3 | 53.7 | | 62.3 | 37.7 | |
| Central | 38.3 | 61.7 | | 52.7 | 47.3 | |
| East | 43.9 | 56.1 | | 58.1 | 41.9 | |
| North-east | 43.3 | 56.7 | | 55.9 | 44.1 | |
| West | 56.6 | 43.4 | | 74.7 | 25.3 | |
| South | 77.9 | 22.1 | | 72.2 | 27.8 | |
| **Education** | | | <0.001 | | | <0.001 |
| No education | 41.5 | 58.5 | | 41.5 | 58.5 | |
| Up to primary | 46.4 | 53.6 | | 50.8 | 49.2 | |
| Up to secondary | 53.6 | 46.4 | | 64.8 | 35.2 | |
| More than secondary | 65.1 | 34.9 | | 76.3 | 23.7 | |
| **Occupation** | | | <0.001 | | | <0.001 |
| Govt./ Non-Govt. employee | 60.7 | 39.3 | | 75.5 | 24.5 | |
| Daily wage/ causal labourer | 46.8 | 53.2 | | 46.6 | 53.4 | |
| Self employed | 49.1 | 50.9 | | 58.2 | 41.8 | |
| Homemaker | 59.6 | 40.4 | | 84.0 | 16.0 | |
| Student | 50.5 | 49.5 | | 52.8 | 47.2 | |
| Others | 50.1 | 49.9 | | 58.0 | 42.0 | |
| **Wealth quintile** | | | <0.001 | | | <0.001 |
| $Q_1$ (poorest) | 39.1 | 60.9 | | 46.1 | 53.9 | |

*(Continued)*

**Table 2.** (Continued)

| Characteristics | Smoke-free status in | | | | | |
| --- | --- | --- | --- | --- | --- | --- |
| | Home (n = 15,254) | | | Workplace (n = 71,046) | | |
| | Yes % | No, % | p-value | Yes, % | No, % | p-value[1] |
| $Q_2$ | 48.7 | 51.3 | | 56.7 | 43.3 | |
| $Q_3$ | 54.5 | 45.5 | | 62.1 | 37.9 | |
| $Q_4$ | 62.8 | 37.2 | | 74.2 | 25.8 | |
| $Q_5$ (richest) | 59.6 | 40.4 | | 72.5 | 27.5 | |
| **Knowledge status about children' illness** | | | <0.001 | | | 0.484 |
| Yes | 52.1 | 47.9 | | 62.9 | 37.1 | |
| No | 43.4 | 56.6 | | 60.0 | 40.0 | |

Q1: Poorest; Q2: Poorer; Q3: Middle; Q4: Richer; Q5: Richest.

their homes and workplace. The major proportion of participants who lived in rural areas was reported to follow smoke-free status both in their homes and workplace. The Chi-square results revealed that all factors except respondents' age and gender were significantly associated with the home-based smoke-free status, whereas all other factors except age were significantly associated with smoke-free status at the workplace (p<0.001) (See Table 2).

### 3.2 Factors associated with smoke-free practices at home and workplace

Only significant factors were considered to perform the logistic regression model for identifying the state of smoke-free status at home and workplaces. Since the VIF of all selected independent variables was less than 10, it indicated that there were no multicollinearity problems among the selected independent variables (Table 3). We found that non-smokers were more likely to live and work in a completely smoke-free environment in their home [AOR = 3.29, 95% CI = 3.00 to 3.59, p<0.001] and workplace [AOR = 1.63, 95% CI = 1.40 to 1.89, p<0.001] compared to smokers (Table 3). We also found that female participants were more likely to enjoy the smoke-free environment at their workplace. Other religious people were 1.94 [AOR = 1.94, 95% CI = 1.65 to 2.28, p<0.001] times more likely to follow a complete smoke-free policy at home compared to Hindu religious people. The participants who lived in rural areas were 1.38 [AOR = 1.38, 95% CI = 1.25 to 1.51, p<0.001] and 1.25 [AOR = 1.25, 95% CI = 1.04 to 1.48, p = 0.015] times more likely to live in a completely smoke-free environment at home and workplace, respectively compared to urban counterparts. The participants who lived in the South region of India had the highest tendency to continue living [AOR = 5.34,95% CI = 4.76 to 5.99, p<0.001]and working [AOR = 1.84,95% CI = 1.48 to 2.29, p<0.001] with complete smoke-free policy compared to other regions. Highly educated participants had a higher tendency to maintain a completely smoke-free environment at their home [AOR = 1.88, 95% CI = 1.69 to 2.08, p<0.001] and workplace [AOR = 2.31, 95% CI = 1.83, 2.93, p<0.001]. Students [AOR = 0.82, 95% CI = 0.73 to 0.92, p = 0.001]and other participants [AOR = 0.86,95% CI = 0.75 to 0.98, p<0.001]had the lowest tendencies to live in a completely smoke-free environment at their home, while daily wise labor [AOR = 0.47,95% CI = 0.38 to 0.57, p<0.001], self-employed [AOR = 0.56,95% CI = 0.46 to 0.68, p<0.001], and students [AOR = 0.49,95% CI = 0.33 to 0.74, p = 0.001] were less likely to report that their workplace was smoke-free compared to government/non-government employees. The participants who belong to rich families were 1.18 [AOR = 1.18, 95% CI = 1.03 to 1.35, p = 0.018] times more likely to live completely smoke-free in their homes. Whereas the rich [AOR = 1.38, 95% CI = 1.06 to 1.79, p = 0.016] and richest [AOR = 1.42, s95% CI = 1.10 to 1.82, p = 0.007] participants had a higher tendency to report

**Table 3. Multivariable logistic regression results.**

| Variable | | Smoke-free status in | | | | | |
|---|---|---|---|---|---|---|---|
| | | Home | | | Workshop | | |
| | | AOR (95% CI of AOR) | p-value | VIF | AOR (95% CI of AOR) | p-value | VIF |
| **Tobacco smoking** | Smoker | 1.00 | | 1.05 | 1.00 | | 1.11 |
| **Status** | Non-smoker | 3.29 (3.00,3.59) | <0.001 | | 1.63 (1.40, 1.89) | <0.001 | |
| **Gender** | Male | - | | | 1 | | 1.11 |
| | Female | - | | | 1.23 (1.00, 1.51) | 0.048 | |
| **Marital status** | Single | 1.00 | | 1.13 | 1.00 | | 1.06 |
| | Married | 0.96 (0.87, 1.06) | 0.420 | | 0.95 (0.79, 1.14) | 0.604 | |
| | Separated/others | 1.08 (0.94, 1.24) | 0.274 | | 0.98 (0.68, 1.43) | 0.934 | |
| **Religion** | Hindhu | 1.00 | | 1.01 | 1.00 | | 1.01 |
| | Muslim | 0.89 (0.77, 1.01) | 0.074 | | 1.07 (0.88, 1.31) | 0.510 | |
| | Christian | 0.99 (0.81, 1.21) | 0.901 | | 0.71 (0.48, 1.05) | 0.084 | |
| | Others | 1.94 (1.65, 2.28) | <0.001 | | 1.09 (0.78, 1.55) | 0.589 | |
| **Place of residence** | Urban | 1.00 | | 1.21 | 1.00 | | 1.19 |
| | Rural | 1.38 (1.25, 1.51) | <0.001 | | 1.25 (1.04, 1.48) | 0.015 | |
| **Region** | North | 1.00 | | 1.01 | 1.00 | | 1.01 |
| | Central | 0.95 (0.84, 1.08) | 0.434 | | 0.87 (0.71, 1.06) | 0.170 | |
| | East | 1.25 (1.11, 1.41) | <0.001 | | 1.42 (1.13, 1.79) | 0.003 | |
| | North-east | 1.26 (1.09, 1.4) | 0.001 | | 1.18 (0.94, 1.49) | 0.162 | |
| | West | 1.64 (1.42, 1.89) | <0.001 | | 1.67 (1.25, 2.24) | 0.001 | |
| | South | 5.34 (4.76, 5.99) | <0.001 | | 1.84 (1.48, 2.29) | <0.001 | |
| **Education** | No education | 1.00 | | 1.44 | 1.00 | | 1.46 |
| | Up to primary | 1.13 (1.04, 1.22) | 0.004 | | 1.36 (1.12, 1.64) | 0.001 | |
| | Up to secondary | 1.40 (1.29, 1.53) | <0.001 | | 1.83 (0.50, 2.24) | <0.001 | |
| | More than secondary | 1.88 (1.69, 2.08) | <0.001 | | 2.32 (1.83, 2.93) | <0.001 | |
| **Occupation** | Govt./Non-Govt. employee | 1.00 | | 1.07 | 1.00 | | 1.10 |
| | Daily wise/causal laborer | 0.85 (0.76, 0.96) | 0.008 | | 0.47 (0.38, 0.57) | <0.001 | |
| | Self-employed | 0.91 (0.82, 1.02) | 0.094 | | 0.56 (0.46, 0.68) | <0.001 | |
| | Homemaker | 1.05 (0.91, 1.21) | 0.530 | | 2.06 (1.33, 3.21) | 0.001 | |
| | Student | 0.82 (0.73, 0.92) | 0.001 | | 0.49 (0.33, 0.74) | 0.001 | |
| | Others | 0.86 (0.75, 0.98) | 0.025 | | 0.63 (0.37, 1.08) | 0.095 | |
| **Wealth quintile** | $Q_1$(poorest) | 1.00 | | 1.41 | 1.00 | | 1.47 |
| | $Q_2$ | 0.98 (0.89, 1.07) | 0.623 | | 1.17 (0.96, 1.44) | 0.121 | |
| | $Q_3$ | 1.08 (0.97, 1.19) | 0.170 | | 1.25 (0.99, 1.56) | 0.051 | |
| | $Q_4$ | 1.18 (1.03, 1.35) | 0.018 | | 1.38 (1.06, 1.79) | 0.016 | |
| | $Q_5$(richest) | 1.10 (0.98, 1.24) | 0.108 | | 1.42 (1.10, 1.82) | 0.007 | |
| **Knowledge status about children' illness** | No | 1.00 | | 1.02 | | | |
| | Yes | 1.22 (1.07, 1.39) | 0.003 | | - | | |

®: Reference category; AOR: Adjusted odds ratio; CI: Confidence interval; SE: Standard error, -indicates that the variable is insignificant in bivariate analysis.

that their workplaces were completely smoke-free compared to the poorest participants. The participants who were knowledgeable and careful about children's illnesses [AOR = 1.22, 95% CI = 1.07 to 1.39, p = 0.003] from other people's smoke had more tendencies to maintain and ensure a complete smoke-free policy in their homes compared to their counterparts.

**Table 4. Correlations between smoke-free status of the home and workplace and SHS exposure (N = 30 State + 2 Unions = 32).**

|  | Smoke-free status at home | Smoke-free status at workplace | SHS exposure at home | SHS exposure at workplace |
|---|---|---|---|---|
| Smoke-free s at home | 1.00 | 0.71** | -0.95** | -0.57** |
| Smoke-free status at the workplace |  | 1.00 | -0.69** | -.69** |
| SHS exposure at home |  |  | 1.00 | 0.60** |
| SHS exposure inworkplace |  |  |  | 1.00 |

SHS-Secondhand smoke; **Correlation is significant at the 0.01 level; *Correlation is significant at the 0.05 level.

## 3.3 Correlation analysis

We performed a correlation analysis to assess the state-level relationship between a smoke-free policy with SHS exposure at home and the workplace. It observed that the smoke-free status of both home and workplace were significantly negatively associated with SHS exposure (Table 4).

## 4 Discussion

The results of the research paper demonstrated that non-smoker participants were more likely to live both completely smoke-free in their homes and workplaces. Based on gender, women's likelihood to live in a smoke-free environment at the workplace was higher than men's [18,19,21,22]. Because of psychological and cultural differences, females were more intended to live in a smoke-free zone in their workplace than their opposite gender [19–22]. Some literature opposed this statement revealing that women's possibility to live in a completely smoke-free environment was less than their male counterparts [20,23–25]. This phenomenon pointed to the fact that in the societal context, mostly men were associated with smoking and women were not even in a position to initiate a protest against this behavior [23,26].

Religious factors were found to be reflected in ensuring smoke-free status. Other religious people were more likely to maintain a smoke-free environment in their homes compared to Hindu religious people. This might be because they are not enough concerned about their family health issues [27]. Religious services both on social and personal levels urged people to not be accustomed to smoking or at least consume fewer cigarettes than non-religious people [28]. People living in rural areas were more prone to live in a completely smoke-free environment both at home and workplace compared to the urban population [18,29,30] which scenario was true in the context of some countries [18,19]. This finding is highly relevant in a country like India where usually rural people are socioeconomically disadvantaged and barely perceived smoking as a risk factor for health [31]. Existing literature agreed with the current study in the sense that on a regional basis, south Indian people possessed the highest tendency to live in a completely smoke-free environment at home and workplace [18,20]. People from central, east, and west were also expected to live in a smoke-free environment in their homes. Alongside the people of the east, the west had a higher opportunity to work in a smoke-free place/zone. The socioeconomic and geographic disparities and diversity in provincial regulations and policies could be responsible for such variation [32].

Literature supported the finding of this study that highly educated people were more likely to live in a smoke-free home environment [18,19,21–24,30]. Educated people were highly aware of the benefits of smoke-free environments to maintaining good health and they instinctively support the smoke-free status [20,21]. However, some studies pointed out contradictory information that the possibility of higher-educated people being in a workplace exposed to

smoking was greater than relatively lower-educated people [19,20]. The violation of smoke-free status at the workplace was high among day workers, students and self-employed people. Both the daily/casual laborers and students had lower chances to live in a completely smoke-free environment at home. The poor implication of campus smoking behavior and lack of social norms could be possible reasons for this. This might be a strong indication of poor implementation of COTPA [33]. Previous literature suggested the fact that people of other occupations tended to experience a lower risk to have an unspoiled smoking contaminated job place as well as a home environment than people engaged in other government / non-government professional careers [20,30].

People who possessed a relatively high level of income were more prone to have a completely smoke-free household and workplace than the poorest ones [21,24,34]. Therefore, socioeconomic conditions contributed to enhancing the cautiousness among the people. Some previous reports demonstrated that tobacco addiction could exacerbate and lock people into poverty. However, due to financial hardship poor people were accustomed to tobacco as a relief from their stress. It could create a higher level of addiction and might become a greater barrier to quitting smoking [35]. If a child belonged to a well-educated and high-income family, they could have a chance to live in a smoke-free better environment which would decrease the exposure risk both at home and in the workplace [21]. The research paper demonstrated that if a parent was a regular smoker both at home and workplace then the chance of children being exposed to SHS would increase [20]. It could be detrimental to children in a two-fold way. They could fall victim to various illnesses and along with that, the risk of smoking could also increase among teenagers if their parents were already accustomed to it [34,36]. Parents having this knowledge intend to prefer a smoke-free policy both at their home.

This study also observed that smoke-free status in the home and the workplace were significantly positively associated with each other whereas both policies were significantly negatively associated with different settings of secondhand smokers. It was indicated that smoke-free status played a vital role in controlling secondhand smoking-induced problems. However, enacting a policy does not necessarily mean that it would be completely implemented throughout the country and this is true for the COTPA as well. For instance, in 2020, a study was conducted in Delhi assessing the implementation level of COTPA 2003 in 376 public places in the city. Various sections of the act such as section 4 (smoking in public places-59.28%), 6-a (implementation level-68.57%), and 6-b (implementation level-52.85%) were observed to be violated to some extent [37]. The case of Bengaluru was also similar in the sense that this city also witnessed the violation of various sections of the act. The absence of a 'non-smoking sign' was 89.7% and 76% of the educational establishments had an abundance of shops selling nicotine products closer to their radius [38]. Besides the cities, the rural and tribal populations also suffer from the problem of the lack of implementation of smoke-free status. India possesses a tribal population of more than 100 million who also belong to the highest level of smoking and because of location and associated resource deprivation, implementation of policies is poor among them [39].

However, it is prevalent from the previous discussion that some sub-groups of the respondents are more exposed to secondhand smoke than their counterparts. While looking for a causal relationship, the literature points out the fact that if workplaces put an embargo on smoking, it highly decreases the rate of smoking and as a result, it decreases the exposure to SHS in the workplace also [40]. Furthermore, if an individual is forced to refrain from smoking in the workplace, it would ultimately lead to quitting smoking at home also. Studies showed that banning any kind of advertisement of cigarettes in the media or the inclusion of smoking in movie scenes will greatly curb people's desire to smoke [41]. Now, as higher educated people and people with higher income levels are generally more exposed to such workplace bans as

well as awareness-raising campaigns against SHS, their rate of engaging in smoking and exposure to SHS is relatively lower.

Though the existing literature showed discrepancies between the enacting and implementation of a policy, it was also mentioned that a collaborative emphasis on the implementation of these policies from the institutions, policymakers, academics, media, and civil society would increase the level of implementation [38,42]. The literature also mentioned that studying the patterns of smoking from different angles will help further improve preventive tools for smoke-free policies [39]. The current study also made a similar observation that smoke-free status and SHS exposure were negatively correlated. It is indicated that an improvement in the smoke-free status could be crucial in reducing the proportion of SHS at every level.

## 4.1 Strength and limitations

This study was conducted based on a large national representative GATS survey. This survey was considered a global standard for monitoring the impact of key tobacco control policies. Earlier studies had analyzed comprehensive information on tobacco smoke-free status at both home and workplace in different countries, but none of them investigated the recent situation in India despite being at a state of high risk for tobacco. Therefore, this research work is expected to assist policymakers in implying more efficient policies for tobacco control in India. In this study, the most significant variable wealth index was indirectly formed. Also, the trend of smoke-free status at home, as well as in the workplace, wasn't explored. However, we didn't adjust some other possible confounding factors which could be explored further (Knowledge, attitude and perception of health issues that smoking causes, cessation, the impact of advertising or counter advertising about smoking in media and so on). Also, it is difficult to handle cross-sectional study as it's unable to make a causal inference and responsible for biases like recall and social desirability.

## 5 Conclusion

This study pointed out some key factors like tobacco smoking status, place of residence, region, education, occupation, wealth quintile, and knowledge status about the children's illness (from other people's smoking) that were significantly associated with complete smoke-free status at home, while all these significant factors along with gender were associated with the complete smoke-free status for the workplace in India. It also focused on the negative association between smoke-free status and SHS exposure. This study aimed to help the policymakers to seek multisectoral, and community-based interventions for smoking in India for reducing the social acceptance of smoking and strongly maintaining smoke-free status both at home and in public workplaces. Indian authorities should emphasize these findings as a supporting document that will help to strengthen the policy implementation of anti-tobacco legislation across the country.

## Acknowledgments

The authors would like to acknowledge the authority of the GATS2-2016/2017.

## Author Contributions

**Conceptualization:** Most. Farida Khatun, Ashis Talukder, Md. Maniruzzaman, Sharlene Alauddin.

**Data curation:** Most. Farida Khatun, Sharlene Alauddin.

**Formal analysis:** Mohammad Ali, Tasnuva Yasmin, Sharlene Alauddin.

**Investigation:** Md. Maniruzzaman, Sharlene Alauddin.

**Methodology:** Mohammad Ali, Most. Farida Khatun.

**Project administration:** Tasnuva Yasmin.

**Resources:** Tasnuva Yasmin.

**Supervision:** Mohammad Ali.

**Validation:** Most. Farida Khatun.

**Visualization:** Tasnuva Yasmin, Ashis Talukder, Md. Maniruzzaman.

**Writing – original draft:** Mohammad Ali, Sharlene Alauddin.

**Writing – review & editing:** Mohammad Ali, Ashis Talukder, Md. Maniruzzaman, Sharlene Alauddin.

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
