## [Decision Letter · Decision Letter 0]

26 Feb 2021

PONE-D-20-41034

Smoke-free policies in the home and in the workplace among Indian people: Evidence from Global Adult Tobacco Survey Data-2017

PLOS ONE

Dear Dr. Alauddin,

Thank you for submitting your manuscript to PLOS ONE. After careful consideration, we feel that it has merit but does not fully meet PLOS ONE’s publication criteria as it currently stands. Therefore, we invite you to submit a revised version of the manuscript that addresses the points raised during the review process.

We look forward to receiving your revised manuscript.

Kind regards,

Stanton A. Glantz

Academic Editor

PLOS ONE

Journal Requirements:

"No"

6. We note you have included a table to which you do not refer in the text of your manuscript. Please ensure that you refer to Table 2 in your text; if accepted, production will need this reference to link the reader to the Table.

Reviewers' comments:

Reviewer's Responses to Questions

**Comments to the Author**

1. Is the manuscript technically sound, and do the data support the conclusions?

Reviewer #1: Yes

Reviewer #2: Yes

2. Has the statistical analysis been performed appropriately and rigorously? 

Reviewer #1: Yes

Reviewer #2: No

3. Have the authors made all data underlying the findings in their manuscript fully available?

Reviewer #1: Yes

Reviewer #2: No

4. Is the manuscript presented in an intelligible fashion and written in standard English?

Reviewer #1: No

Reviewer #2: No

5. Review Comments to the Author

Reviewer #1: I have read this paper with interest. I have some observations. There have been some recent (2020) publications looking at SHS exposure at homes and workplaces. These have been listed below for the authors. Given that, the authors need to better explain the rationale for this study. Substantial edits are required throughout the manuscript (see below). The manuscript requires a thorough copy-editing for English language, structure and meaning that is conveyed.

Abstract

The authors say “and determine and determine the association between and within smoke-free policies at different setting where secondhand smoking occurs.” This is not very clear. This could be reworded to make this part of the objective clearer.

The introductory sentences in the abstract could be removed and more detail could be provided on data source and methods.

Description of the results is unclear and could be clarified. The authors mention factors correlated with complete smoke-free policies at home and at workplace and subsequent sentence says therefore both were significantly associated. This is unclear. The authors also mention that both policies were significantly negatively associated with different settings of secondhand smokers. This part “different settings” also should be clarified. What are these different settings? What analysis was done.

Overall, abstract requires to be reworked significantly to make objectives, data sources, methods, results and conclusion/policy implication clearer.

Introduction

Page 3

- National Tobacco Control Programme was established in 2007-08.

- Smoking zones are allowed only in hotels with more than 30 rooms, restaurants with more than 30 seats and airports.

- ‘Open spaces’ could be reworded as several open spaces are covered e.g. bus-stops, stadiums etc.

- Reference/s should be provided for “some previous country- or community-based researches shown that the presence of policies that restrict smoking in the workplace and home may vary by geographical region, occupation, industry and socioeconomic status.” And if this is in the context of GATS, what does ‘industry’ mean here?

- Not able to understand here the way authors are trying to justify the importance of studying context of implementation of policies. Yes, the level of implementation may vary by geographic region (and GATS does not measure level of implementation). I believe, at best, the authors are mentioning about correlates of smoke-free policies and the variation therein (between urban/rural geographic regions, across occupational of SES categories). The rationale for studying associations at state-level also needs proper justification. So current justification here needs to be revised.

- The authors may also want to have a look at these published papers to clarify their justification for this study:

1) https://pdfs.semanticscholar.org/870c/449e35399633710a9ad4b223e46947102af1.pdf?_ga=2.118417226.2146522655.1614069121-810001830.1607255405

2) https://link.springer.com/article/10.1007/s11356-019-07341-x

3) https://link.springer.com/article/10.1007/s11356-020-10107-5

Methods

Page 5, section 2.2

Why only knowledge about “children’s illness” would be included as an independent variable? SHS exposure causes illness among all.

Authors could provide more information about independent variables and groupings. The authors just mention ‘household assets’ but how was SES variable derived from that? Perhaps a supplementary table with variable definitions could be useful.

Section 2.3 (Data analysis) – The authors used logistic regression to study the correlates of living in smoke-free homes and working in smoke-free workplaces. Some language edits are required here for clarity. Did the authors perform any multi-collinearity diagnostics? It should be reported. The analysis for state-level within and between analysis needs further description for clarity. This seems relevant because there is no description of how the “State” variable was treated in the analysis. From the results presented (Table 3 and Correlation matrix), it seems only overall National level estimates have been used and presented.

Results

In tables 1 and 2 the authors could club the other three categories for both outcomes unless they are going into their details. In Table 3, why are gender (for SF at home) and age groups (for SF at workplace) missing as In Table 1 gender is significantly associated with the outcome and in Table 2, age group is also significantly associated (p<0.05). Authors should check this.

Ok. At the end of results section and after seeing the correlation matrix, one realizes that it is not the “different settings of second hand smokers” but it is SHS exposure (at home, workplace and public places). This needs to be clarified in the methods and also in the abstract. “Different settings of secondhand smokers” is confusing. Again, when SHS exposure at public places has not been discussed throughout the paper, what is the rationale for bringing that in in the correlation matrix?

Discussion

Page 7

- First line of discussion – Non-smoker participants were more likely (NOT priorities)….

- Well, most of the results and discussion deals with for e.g. females vs. males are more likely to live in smoke-free homes/workplaces and similar findings. What is not discussed in Discussion section, is the policy implications of the findings. In India, the smoke-free legislation is a national law (under COTPA 2003) and probably the level of implementation varies across geographic areas (rural/urban, national regions, states). So how are the findings specifically relevant to the national policy scenario? That needs to be discussed in detail.

Reviewer #2: This study used data from the 2016/17 wave of the GATS conducted in India to analyze the prevalence and determinants of smoke-free policies in the home and in the workplace among Indian people aged 15+. Towards this research aim, this study did a great job to perform the analyses rigorously and interpret the results in great detail. However, it is less clear what the second aim -- assessing the state-level within and between relationships among the nature of smoke free policies prevailing at different settings where secondhand smoking occurs -- is about, and why it is important to examine this aim.

1. Title: 2017 should be changed to 2016/2017 because the GATS2-India was conducted in the period between August 2017 and February 2017.

2. Abstract lines 5-6: Why is it necessary to examine the correlation between smoke-free policies in the home and smoke-free policies in the workplace?

3. There is a lack of explanation about the secondhand smoke measure. In the abstract (page 2, lines 6-7), Introduction section (page 4, lines 8-9), Statistical Analysis section (page 5, lines 6-8 of the second paragraph), and Results section (page 7, the second paragraph), the authors talked about the associations between smoke-free policies and different settings of secondhand smoke. However, there is no explanation about the definition of secondhand smoke, how many settings were examined, and how each setting of secondhand smoke was measured. Also, is secondhand smoke an outcome variable or independent variable? This information needs to be described in the Methods section.

4. Introduction, 1st paragraph, lines 9-10: “Among the total tobacco consumers, 42.4% are male and 14.2% are female.” This sentence does not look right because the male tobacco consumers (42.4%) and female consumers (14.2%) should add up to total tobacco consumers (100%). Does “42,2%” refer to “the prevalence of tobacco use”?

5. Introduction, 2nd paragraph, lines 10-12: For the sentence which begins with “Some previous country or community ….”, can you cite some references to justify this sentence?

6. Sample size: Page 4, lines 11-12 of the second paragraph: “After removing respondents with missing values ...” � Please explain the reason for missing values. For example, how many respondents were excluded due to missing values for the outcome variable, and how many due to missing values for independent variables?

7. The paragraph about the independent variables need to be substantially expanded by providing detailed definition of each independent variable. For example, tobacco smoking status was classified into smokers, and non-smokers. How do you define “smokers” and “non-smokers defined”?

8. More explanation is needed about what analyses utilized the chi-square test (which was stated in the “Statistical Analysis” paragraph). Also, please clarify meaning of “a feature selection method for further analysis”.

9. The classification of the two outcome variables – smoke-free policy at home, and smoke-free policy in the workplace – seems confusing and inconsistent. First, In the “Variable of the study” paragraph, the smoke-free policy at home variable was classified as a binary variable: “living in a complete smoke-free home” vs. “not living in a complete smoke-free home“. Second, in the Results section, this variable was classified into a 3-category variable: “complete smoke-free home”, “partial smoke-free home”, and “no rules about smoking”. Third, in Table 1, this variable was classified into a 4-cateogy variable: “never allowed”, “not allowed, but exception”, “no rules”, and “allowed”. It is not clear whether the chi-square test was conducted using the first classification, or the second classification, or the third classification. Similar problem also exists for the smoke-free policy in the workplace variable.

10. Page 6, lines 9-10 of the second paragraph: To justify the claim that “Participants who lived in South region … had the highest tendency to live …..compared to other regions.”, the multivariate logistic regression model needs to be revised using “south” as the reference group.

 

12. The first sentence in the Conclusion section indicates that “54.4% had complete smoke-free polices at home”, while page 6, line 1 says that 50.4% had complete smoke-free policies at home. Which one is correct?

13. Tables 1-2 need to show total sample size, change “tobacco polices” to “smoke-free policies” in the title, and add footnote to explain what the superscripts “1”, “(a)”, and “(b)” mean.

14. Table 3 needs to add outcome variable information in the title, and change the term “odds ration” to “adjusted odds ratio”.

6. PLOS authors have the option to publish the peer review history of their article (what does this mean?). If published, this will include your full peer review and any attached files.

Reviewer #1: **Yes: **Dr. Gaurang P. Nazar

Reviewer #2: No

---

## [Author Response · Author response to Decision Letter 0]

5 May 2021

Authors’ Response to Reviewers

Manuscript ID: PONE-D-20-41034

Title: Smoke-free policies in the home and in the workplace among Indian people: Evidence from Global Adult Tobacco Survey Data-2017

Dear Respected Editor-in-Chief,

First of all, we thank the reviewer and the editor for their useful suggestions for improving the manuscript. We have now revised the manuscript accordingly. Thank you once again and looking forward to the final acceptance in your esteemed journal.

Reviewer #1:

Reviewer#1-1: I have read this paper with interest. I have some observations. There have been some recent (2020) publications looking at SHS exposure at homes and workplaces. These have been listed below for the authors. Given that, the authors need to better explain the rationale for this study. Substantial edits are required throughout the manuscript (see below). The manuscript requires a thorough copy-editing for English language, structure and meaning that is conveyed. 

Authors: Thank you for your suggestions. The manuscript has been revised and modified by considering your instruction. 

Reviewer#1-2: In Abstract, the authors say “and determine and determine the association between and within smoke-free policies at different setting where secondhand smoking occurs.” This is not very clear. This could be reworded to make this part of the objective clearer. 

Authors: Thank you very much for asking this question. We are totally agreed with this reviewer. We update the objectives in the revised manuscript as follows: 

Old objective: To determine the correlation between smoke-free policies in the home and workplace and determine the association between and within smoke-free policies at different setting where secondhand smoking occurs

New Objective: To determine the correlates of smoke-free policies in the home and workplace along with assessing the relationship between smoke-free policies and secondhand smoker’s exposure.

[Revised Manuscript Page #2; Abstract: Line#: 1-3]

Reviewer#1-3: The introductory sentences in the abstract could be removed and more detail could be provided on data source and methods. 

Authors: Thank you very much for asking this question. We are already removed the introductory sentences are removed from abstract and added the data source and methods more details in the abstract in the revised manuscript. Thanks again. 

[Revised Manuscript Page #2]

Reviewer#1-4: Description of the results is unclear and could be clarified. The authors mention a factor correlated with complete smoke-free policies at home and at workplace and subsequent sentence says therefore both were significantly associated. This is unclear. The authors also mention that both policies were significantly negatively associated with different settings of secondhand smokers. This part “different settings” also should be clarified. What are these different settings? What analysis was done? 

Authors: Thank you very much for asking this question. Actually, this study presents a comparative study of completely smoke-free policy between home and workplace. This study also assess the relationship between smoke-free polices and secondhand smokers. To fill-up the objectives, we have implemented logistic regression model to determine the correlates of smoke-free policies in the home and workplace. We have also implemented Pearson correlation co-efficient to explore the relationship between smoke-free policies and proportion of secondhand smoker’s exposure at both home and workplace. We are updated the abstract in details in the revised manuscript. Thanks again.

Old abstract: Tobacco smoking is one of the biggest global health concerns of this century. Secondhand smoking has a significant contribution to the increasing burden of cancers, chronic diseases and associated mortality. India is the second largest consumer of tobacco after China in the world. A number of researches were conducted in India related to tobacco for improving tobacco control policies. However, this is the first study to determine the correlation between smoke-free policies in the home and workplace and determine the association between and within smoke-free policies at different setting where secondhand smoking occurs. Results found that tobacco smoking status, place of residence, geographical regions, education, employment status, household asset and knowledge status about children being affected the rate of second hand smoking, and were significantly correlated at p < 0.001 with the complete smoke-free policies both in the home and in the workplace in India. Therefore, smoke free policies in the home and in the workplace were significantly positively associated with each other whereas both policies were significantly negatively associated with different settings of secondhand smokers.

New abstract: 

Aim: The aim of this study was to determine the correlates of smoke-free policies in the home and workplace along with assessing the relationship between smoke-free policies and secondhand smoker’s exposure.

Methods: Data was collected from Global Adult Tobacco Survey Data (GATS)-2017. It was a household survey that included the people aged 15 years or older and covered all the 30 states and 2 Union Territories (UTs) in India. Logistic Regression model was used to determine the correlates of smoke-free policies in the home and workplace. Pearson correlation was also used to explore the relationship between smoke-free policies and proportion of secondhand smoker’s exposure at both home and workplace. 

Results: Results found that tobacco smoking status, place of residence, geographical regions, education, employment status, household asset, and knowledge status about the children’s illness from other people’s smoking were significantly correlated at p<0.001 with the complete smoke-free policies both in home and workplace in India. The study also found that smoke-free policies in both home and workplace were significantly negatively associated with secondhand smoker’s exposure. 

Conclusion: This study is expected to help the policymakers to improve smoke-free policies and ensure a better environment both in home and workplace.

[Revised Manuscript Page #2]

Reviewer#1-5: Overall, abstract requires to be reworked significantly to make objectives, data sources, methods, results and conclusion/policy implication clearer. 

Authors: Thank you very much for asking this question and give us an opportunity to improve the revised manuscript. We are totally agreed with this reviewer. We are already added in details of the objectives, data sources, results and conclusion/policy implication in the abstract of the revised manuscript as follows: 

“Aim: The aim of this study was to determine the correlates of smoke-free policies in the home and workplace along with assessing the relationship between smoke-free policies and secondhand smoker’s exposure.

Methods: Data was collected from Global Adult Tobacco Survey Data (GATS)-2017. It was a household survey that included the people aged 15 years or older and covered all the 30 states and 2 Union Territories (UTs) in India. Logistic Regression model was used to determine the correlates of smoke-free policies in the home and workplace. Pearson correlation was also used to explore the relationship between smoke-free policies and proportion of secondhand smoker’s exposure at both home and workplace. 

Results: Results found that tobacco smoking status, place of residence, geographical regions, education, employment status, household asset, and knowledge status about the children’s illness from other people’s smoking were significantly correlated at p<0.001 with the complete smoke-free policies both in home and workplace in India. The study also found that smoke-free policies in both home and workplace were significantly negatively associated with secondhand smoker’s exposure. 

Conclusion: This study is expected to help the policymakers to improve smoke-free policies and ensure a better environment both in home and workplace.”

[Revised Manuscript Page #2]

Thank again for giving us an opportunity to improve the revised manuscript.

Reviewer#1-6: Introduction, Page 3: National Tobacco Control Programme was established in 2007-08.

Authors: Thank again this reviewer for giving us an opportunity to improve the revised manuscript. We are totally agreed with this reviewer that National Tobacco Control Programme was established in 2007-08. We fix it and update the revised manuscript. 

[Revised Manuscript Page #3, Section #1]

Reviewer#1-7: Introduction, Page 3: Smoking zones are allowed only in hotels with more than 30 rooms, restaurants with more than 30 seats and airports. 

Authors: Thank again this reviewer for giving us an opportunity to improve the revised manuscript. According to your overall suggestions, we exclude this statement from our revised manuscript to rearrange the introduction. Thanks again. 

Reviewer#1-8: Introduction, Page 3: ‘Open spaces’ could be reworded as several open spaces are covered e.g. bus-stops, stadiums etc. 

Authors: Thank again this reviewer for giving us an opportunity to improve the revised manuscript. According to your overall suggestions, we exclude this statement from our revised manuscript to rearrange the introduction. Thanks again.

Reviewer#1-9: Introduction, Page 3: Reference/s should be provided for “some previous country- or community-based researches shown that the presence of policies that restrict smoking in the workplace and home may vary by geographical region, occupation, industry and socioeconomic status.” And if this is in the context of GATS, what does ‘industry’ mean here? 

Authors: Thank you for your kind suggestions. Some words are reworded according to your advice. Unfortunately Reference/s for some statements which are written on the basis of some previous evidence based literature was missed. We had included the necessary references in the required places in the revised manuscript. 

In the context of GATS, we are totally agreed with this reviewer. So, we omit the words “industry” from the revised manuscript. Thanks again

Reviewer#1-10: Introduction, Page 3: Not able to understand here the way authors are trying to justify the importance of studying context of implementation of policies. Yes, the level of implementation may vary by geographic region (and GATS does not measure level of implementation). I believe, at best, the authors are mentioning about correlates of smoke-free policies and the variation therein (between urban/rural geographic regions, across occupational of SES categories). The rationale for studying associations at state-level also needs proper justification. So current justification here needs to be revised. The authors may also want to have a look at these published papers to clarify their justification for this study:

1. https://pdfs.semanticscholar.org/870c/449e35399633710a9ad4b223e46947102af1.pdf?_ga=2.118417226.2146522655.1614069121-810001830.1607255405

2. https://link.springer.com/article/10.1007/s11356-019-07341-x

3. https://link.springer.com/article/10.1007/s11356-020-10107-5

Authors: Thank again this reviewer for giving us an opportunity to improve the revised manuscript. It is true that the level of implementation may vary by geographic region and GATS does not measure level of implementation. So according to the suggestion of this reviewer, we emphasis on the correlates of smoke-free policies and demonstrated the variation therein regions (between urban/rural geographic regions, across occupational of SES categories) and already the included the revised manuscript.

Reviewer#1-11: In Method, Page 5, section 2.2: Why only knowledge about “children’s illness” would be included as an independent variable? SHS exposure causes illness among all. 

Authors: Thank again this reviewer for giving us an opportunity to improve the revised manuscript. We not consider other knowledge about SHS exposure causes illness to avoid multi-collinearity problem. 

Reviewer#1-12: In Method, Page 5, section 2.2: Authors could provide more information about independent variables and groupings. The authors just mention ‘household assets’ but how SES variable was derived from that? Perhaps a supplementary table with variable definitions could be useful. 

Authors: Thanks this reviewer for asking this question. We are totally agreed this reviewer. We have already added the information about independent variables along with their groupings of each variable which is presented a supplementary table in the revised manuscript as follows: 

Table. Descriptions of the independent variables along with their categorizations

Factors Descriptions Categorizations

Smoking status Smoking status 1= Smoker, 2= Non-smoker

Age (years) Age of the respondents in years 1=15 to 30 years, 2= 31 to 45 years, 3= 46 to 60 years, 4= Up to 60 years

Gender Gender of respondents 1= Male, 2= Female

Marital status Marital status of respondents 1= single, 2= Married, 3=Separated/others

Religion Religion 1=Hindu,2=Muslim,3=Christian,4= Others

Place of residence Place of residence 1=Urban,2=Rural

Geographical regions Respondent’s geographical regions 1=North, 2= Central, 3=East, 4=North-east, 5=West, 6=South

Education Education status of respondent’s 1=No education, 2=Up to primary, 3=Up to secondary, 4=More than secondary

Occupation Occupation status of the respondents 1=Govt./Non-Govt. employee, 2= Daily wise/causal laborer, 3= Self-employed, 4=Homemaker, 5= Student, 6= Others

Household assets Household assets of the respondents 1=Poorest, 2= Poorer, 3= Middle, 4= Richer, 5= Richest

Knowledgeable of the children Knowledgeable of the children illness 1= No, 2= Yes

[Revised Manuscript Page #17, Supplementation]

The household assets construction procedure is already added in the revised manuscript as follows: 

“In case of household assets, the following indicators were used: whether the household or person living in the household enjoys the facility of i) electricity, ii) electric fan, iii) air conditioner, iv) refrigerator, v) washing machine, vi) radio, vii) television, viii) computer/laptop, ix) internet connection, x) moped/scooter xi) car, xii) flush toilet, xiii) fixed telephone, xiv) cell telephone. Principal component analysis (PCA) had been conducted and this variable was divided into five categories (poorest, poorer, medium, richer, and richest) on the basis of the quintiles of the first PCA scores.”

[Revised Manuscript Page #5, Section #2, Subsection# 2.3]

Reviewer#1-13: Section 2.3 (Data analysis)--The authors used logistic regression to study the correlates of living in smoke-free homes and working in smoke-free workplaces. Some language edits are required here for clarity. Did the authors perform any multi-collinearity diagnostics? It should be reported. 

Authors: Thanks this reviewer for asking this question. At the initial stage we checked the existence of multi-collinearity problem by considering the value of standard error (SE) but didn’t include it in the manuscript. We are extremely sorry for that. Now, we have fixed it and update the revised manuscript. Thanks again. 

[Revised Manuscript Page #5, Section #2, Subsection# 2.4]

Reviewer#1-14: Section 2.3 (Data analysis)--The analysis for state-level within and between analyses needs further description for clarity. This seems relevant because there is no description of how the “State” variable was treated in the analysis. From the results presented (Table 3 and Correlation matrix), it seems only overall National level estimates have been used and presented. 

Authors: Thanks this reviewer for asking this question. We collected the estimated proportion of smoke-free policies and secondhand smoker’s exposure at both home and workplace from the GATS 2 report for each state to measure the correlation between the state-level proportion of smoke-free policies and secondhand smoker’s exposure at both home and workplace.

[Revised Manuscript Page #5, Section #2, Subsection# 2.3]

Reviewer#1-15: Results, in tables 1 and 2 the authors could club the other three categories for both outcomes unless they are going into their details. In Table 3, why are gender (for SF at home) and age groups (for SF at workplace) missing as In Table 1 gender is significantly associated with the outcome and in Table 2, age group is also significantly associated (p<0.05). Authors should check this. 

Authors: Thanks this reviewer for asking this question. We deeply apologize for such mistakes. There was some misinterpretation in that table. We already fixed Tables in the revised manuscript. Thanks again for giving us an opportunity to improve the revised manuscript. 

Reviewer#1-16: Results, Ok. At the end of results section and after seeing the correlation matrix, one realizes that it is not the “different settings of second hand smokers” but it is SHS exposure (at home, workplace and public places). This needs to be clarified in the methods and also in the abstract. “Different settings of secondhand smokers” is confusing. Again, when SHS exposure at public places has not been discussed throughout the paper, what is the rationale for bringing that in the correlation matrix?

Authors: Thanks this reviewer for asking this question. We have modified the word second hand smokers (SHS) to SHS exposure. The SHS exposure at public places was only depicted in the correlation matrix. We are totally agreed with your justification of being irrational for bringing that point and omitted this section from the correlation matrix. Thanks again for giving us an opportunity to improve the revised manuscript. 

Reviewer#1-17: Discussion, Page 7:-First line of discussion–Non-smoker participants was more likely (NOT priorities)….

Author: Thanks this reviewer for asking this question. We are already modified the first line in the revised manuscript. Thanks again. 

Reviewer#1-17: Discussion, Page 7:-Well, most of the results and discussion deals with for e.g. females vs. males are more likely to live in smoke-free homes/workplaces and similar findings. What is not discussed in Discussion section, is the policy implications of the findings. In India, the smoke-free legislation is a national law (under COTPA 2003) and probably the level of implementation varies across geographic areas (rural/urban, national regions, states). So how are the findings specifically relevant to the national policy scenario? That needs to be discussed in detail. 

Author: Thanks this reviewer for asking this question. We are already modified the whole discussion section the revised manuscript. Thanks again. 

[Revised Manuscript Page #7, 8, 9, Section #4]

Dear Reviewer #1

Thank you very much for your kind guidance and support in helping the authors improve the manuscript. 

We are truly grateful for your time and support.

Kind Regards,

The Authors of the manuscript.

Reviewer #2:

 This study used data from the 2016/17 wave of the GATS conducted in India to analyse the prevalence and determinants of smoke-free policies in the home and in the workplace among Indian people aged 15+. Towards this research aim, this study did a great job to perform the analyses rigorously and interpret the results in great detail. However, it is less clear what the second aim -- assessing the state-level within and between relationships among the nature of smoke free policies prevailing at different settings where secondhand smoking occurs -- is about, and why it is important to examine this aim.

Reviewer#2-1: Title: 2017 should be changed to 2016/2017 because the GATS2-India was conducted in the period between August 2017 and February 2017.

Authors: Thank you for your suggestions. The manuscript has been revised and modified by considering your instruction. 

[Revised Manuscript Page #1, 2; Title of the manuscript]

Reviewer#2-2: Abstract lines 5-6: Why is it necessary to examine the correlation between smoke-free policies in the home and smoke-free policies in the workplace?

Authors: Thank you very much for asking this question. We are totally agreed with this reviewer. We update the objectives in the revised manuscript as follows: 

Old objective: To determine the correlation between smoke-free policies in the home and workplace and determine the association between and within smoke-free policies at different setting where secondhand smoking occurs

New Objective: To determine the correlates of smoke-free policies in the home and workplace along with assessing the relationship between smoke-free policies and secondhand smoker’s exposure.

[Revised Manuscript Page #2; Abstract: Line#: 1-3]

Reviewer#2-3: There is a lack of explanation about the secondhand smoke measure. In the abstract (page 2, lines 6-7), Introduction section (page 4, lines 8-9), Statistical Analysis section (page 5, lines 6-8 of the second paragraph), and Results section (page 7, the second paragraph), the authors talked about the associations between smoke-free policies and different settings of secondhand smoke. However, there is no explanation about the definition of secondhand smoke, how many settings were examined, and how each setting of secondhand smoke was measured. Also, is secondhand smoke an outcome variable or independent variable? This information needs to be described in the Methods section.

Authors: Thank you very much for asking this question. Actually, we do not consider secondhand smoker in our main analysis. We collected the estimated proportion of smoke-free policies and secondhand smoker’s exposure at both home and workplace from the GATS 2 report for each state to measure the correlation between the state-level proportion of smoke-free policies and secondhand smoker’s exposure at both home and workplace. We are already updated it’s in the revised manuscript. 

[Revised Manuscript Page #5; Section#2, Subsection# 2.4]

Reviewer#2-4: Introduction, 1st paragraph, lines 9-10: “Among the total tobacco consumers, 42.4% are male and 14.2% are female.” This sentence does not look right because the male tobacco

 consumers (42.4%) and female consumers (14.2%) should add up to total tobacco consumers (100%). Does “42.2%” refer to “the prevalence of tobacco use”?

Authors: Thank you very much for asking this question. Actually, we do not show the proportion/percentages of male and female tobacco consumers. So, it is not mandatory to reach 100% (their sum). Actually, we show the prevalence of male and female tobacco consumer. In the previous manuscript, we have some mistakes to sentence structure. Now, we fixed it and updated the revised manuscript. Thanks again. 

Old: Among the total tobacco consumers, 42.4% are male and 14.2% are female

New: The prevalence of male and female tobacco consumers is 42.4% and 14.2%.

[Revised Manuscript Page #3; Section#1]

Reviewer#2-5: Introduction, 2nd paragraph, lines 10-12: For the sentence which begins with “Some previous country or community, can you cite some references to justify this sentence?

Authors: Thank you very much for asking this question. We are extremely sorry for our mistake. Now, we added some references in the revised manuscript. Thanks again. 

[Revised Manuscript Page #3-4; Section#1]

Reviewer#2-6: Sample size: Page 4, lines 11-12 of the second paragraph: “After removing respondents with missing values” Please explain the reason for missing values. For example, how many respondents were excluded due to missing values for the outcome variable, and how many due to missing values for independent variables?

Authors: Thank you very much for asking this question. In GATS 2, a total of 74,037 individual interviews were completed with an overall response rate of 92.9%. Among the respondent’s, 7 were refused and 244 responded with don’t know about smoke-free policies at home. After excluding these respondents along with 2740 (missing, refused, don’t know etc.) in the selected independent variables, 71, 046 respondents were selected for final analysis in case of smoke-free policies at home.

The analysis of smoking policies at workplace was restricted to the GATS2 respondents (18258) who were working indoors or both indoors and outdoors but outside their home (adults). Among them, 7 respondents are refused and 374 respondent’s responses don’t know about the work place policies. After excluding these respondent’s (381) and 393 missing, refused, don’t know etc.) in the selected independent variables, 17, 484 respondents were selected for final analysis for smoke-free policies at work place. 

Reviewer#2-7: The paragraph about the independent variables needs to be substantially expanded by providing detailed definition of each independent variable. For example, tobacco smoking status was classified into smokers, and non-smokers. How do you define “smokers” and “non-smokers defined”?

Authors: Thank you very much for asking this question. In Gats 2016-2017 questionnaire, there is a question about tobacco smoking: Does the person currently smoke tobacco including bidhis, cigarattes, hukkah, cigars etc.? (i) Yes, (ii) No, (iii) Don’t know, and (iv) Refused. The frequency table as follows: 

 Frequency %

Yes 11767 15.9

No 62109 83.9

Don’t know 160 0.2

Refused 1 0.00

We omit the respondents who refused and replied with don’t know. Thanks again. 

Reviewer#2-8: More explanation is needed about what analyses utilized the chi-square test (which was stated in the “Statistical Analysis” paragraph). Also, please clarify meaning of “a feature selection method for further analysis”.

Authors: Thank you very much for asking this question. We are extremely sorry for mistake. We are already fixed it in the revised manuscript. Thanks. 

Reviewer#2-9. The classification of the two outcome variables – smoke-free policy at home, and smoke-free policy in the workplace – seems confusing and inconsistent. First, In the “Variable of the study” paragraph, the smoke-free policy at home variable was classified as a binary variable: “living in a complete smoke-free home” vs. “not living in a complete smoke-free home“. Second, in the Results section, this variable was classified into a 3-category variable: “complete smoke-free home”, “partial smoke-free home”, and “no rules about smoking”. Third, in Table 1, this variable was classified into a 4-cateogy variable: “never allowed”, “not allowed, but exception”, “no rules”, and “allowed”. It is not clear whether the chi-square test was conducted using the first classification, or the second classification, or the third classification. Similar problem also exists for the smoke-free policy in the workplace variable.

Authors: Thank you very much for asking this question. In our main dataset, smoke-free policies at home and workplace consist of 4 categories. To assess the association between smoke-free policies and independent variables using Chi-square test, we consider four categories. But to determine the correlates of complete smoke-free policies at both home and workplace converted into binary. It is a common case in public health. 

Reviewer#2-10. Page 6, lines 9-10 of the second paragraph: To justify the claim that “Participants who lived in South region … had the highest tendency to live …..compared to other regions.”, the multivariate logistic regression model needs to be revised using “south” as the reference group.

Authors: Thank you very much for asking this question. We are already done in the revised manuscript. 

 

Reviewer#2-10. The first sentence in the Conclusion section indicates that “54.4% had”, while page 6, line 1 says that 50.4% had complete smoke-free policies at home. Which one is correct?

Authors: Thank you very much for asking this question. This was the type of mistake. Actually, 54.4% had complete smoke-free polices at home correct not 50.4%. We are already fixed it and update the revised manuscript. Thanks again. 

Reviewer#2-10. Tables 1-2 need to show total sample size, change “tobacco polices” to “smoke-free policies” in the title, and add footnote to explain what the superscripts “1”, “(a)”, and “(b)” mean.

Authors: Thank you very much for asking this question. We are already fixed it and update the revised manuscript. Thanks again. 

Reviewer#2-10. Table 3 needs to add outcome variable information in the title, and change the term “odds ration” to “adjusted odds ratio”.

Authors: Thank you very much for asking this question. We are already fixed it and update the revised manuscript. Thanks again.

Dear Reviewer #2

Thank you very much for your kind guidance and support in helping the authors improve the manuscript. 

We are truly grateful for your time and support.

Kind Regards,

The Authors of the manuscript.

---

## [Decision Letter · Decision Letter 1]

25 Jun 2021

PONE-D-20-41034R1

Smoke-free policies in the home and in the workplace among Indian people: Evidence from Global Adult Tobacco Survey Data-2016/2017

PLOS ONE

Dear Dr. Alauddin,

Thank you for submitting your manuscript to PLOS ONE. After careful consideration, we feel that it has merit but does not fully meet PLOS ONE’s publication criteria as it currently stands. Therefore, we invite you to submit a revised version of the manuscript that addresses the points raised during the review process.

There are still many serious methodological issues with your paper.  Should you chose to resubmit, please take extra care to ensure that you properly address all the issues the reviewers raised.  I will be sending the revised manuscript back to the reviewers for reassessment.

We look forward to receiving your revised manuscript.

Kind regards,

Stanton A. Glantz, PhD

Academic Editor

PLOS ONE

Reviewers' comments:

Reviewer's Responses to Questions

**Comments to the Author**

1. If the authors have adequately addressed your comments raised in a previous round of review and you feel that this manuscript is now acceptable for publication, you may indicate that here to bypass the “Comments to the Author” section, enter your conflict of interest statement in the “Confidential to Editor” section, and submit your "Accept" recommendation.

Reviewer #1: (No Response)

Reviewer #2: (No Response)

2. Is the manuscript technically sound, and do the data support the conclusions?

Reviewer #1: Yes

Reviewer #2: No

3. Has the statistical analysis been performed appropriately and rigorously? 

Reviewer #1: Yes

Reviewer #2: No

4. Have the authors made all data underlying the findings in their manuscript fully available?

Reviewer #1: No

Reviewer #2: Yes

5. Is the manuscript presented in an intelligible fashion and written in standard English?

Reviewer #1: No

Reviewer #2: No

6. Review Comments to the Author

Reviewer #1: OVERALL

The authors have addressed a number of comments that were raised. The manuscript still requires significant copy-editing. There are some concerns regarding the terminologies used. The authors should ensure correct terms are used which are consistent with previous tobacco literature. There are some minor recommendations that I believe would improve the quality of the paper.

ABSTRACT:

Aim – edit “secondhand smoker’s exposure” to “secondhand smoke exposure”. Please ensure this changed is reflected throughout the manuscript

Methods – edit “proportion of secondhand smoker’s exposure” to “proportion of participants exposed to secondhand smoke”

Results – edit “Results found that” to “We found that”

edit “secondhand smoker’s exposure” to “secondhand smoke exposure”

Conclusion – smokefree policies are already in place in India. Maybe authors could emphasize “comprehensive smokefree policies without any exceptions” and their “enhanced monitoring.”

This could be a part of discussion and conclusion but authors should note - Penalty for violation of smoke-free legislation in public places is currently only INR 200 (approx USD 2.7) which needs to be increased considerably taking into account inflation and rise in purchasing power/incomes. The proposed COTPA Amendment Bill (2020) proposes to increase this penalty to INR 2000 but this Bill has not yet been tabled in the Parliament.

INTRODUCTION

First para – 6th line from bottom – remove “of them”

First para – 5th line from bottom – “…tendency to smoke themselves but exposure to SHS…” can be replaced by “…tendency to smoke, exposure to SHS….”

Second para – last sentence – “a progress of smoke-free policies” should be replaced by “progress in smoke-free policies”. “government still couldn’t reach its ultimate goal” could be replaced by “the policies are still not comprehensive”

Third para – “A number of countries or community-based Researches” could simply be replaced by “Previous research has”. And “secondhand smokers exposure” should be replaced at all places by “exposure to secondhand smoke”

Variables of the study – I think the household asset variable should be termed as the “wealth quintile” or “asset quintile”. This is typically the term used in similar research papers.

Similarly, there is a problem with terminologies here “proportion of smoke-free policies and secondhand smoker’s exposure at both home and workplace”. This could be replaced by “proportion of participants living in smoke-free homes; working in smoke-free workplaces and proportion of participants exposed to secondhand smoke.” The authors should take care of these terminologies throughout the paper. The authors may also want to clarify here whether the proportion of participants exposed to SHS was restricted to non-smokers only.

Statistical analysis – “the multicollinearity problem” could be replaced by just “multicollinearity”. The authors should run their regression models with ‘collin’ command (if using STATA) after running each logistic model and look for higher VIF values. While there is no rule of thumb for this, they could report that multicollinearity was assessed and all VIF values were less than 10, indicating no significant multicollinearity between explanatory variables. Multicollinearity between variables can lead to high SEs of coefficients and there are less chances of them being significant. Hence, only of such variables should be retained in the model.

Apologies for not pointing this out earlier but in the statistical analysis section, the authors should report about using survey weights “svyset” to account for the complex multistaged GATS survey design.

RESULTS

Just want to point out here to the authors that SHS typically stands for “Secondhand Smoke” in tobacco literature and not “Secondhand Smoker’s”. This should be corrected throughout the paper and also in tables and their legends. The correct term should be “Secondhand smoke exposure” or SHS exposure.

CONCLUSION

This needs to be strengthened. As mentioned earlier, the authors studied complete smoke-free policy vs. no/partial smoke-free policy both at home and at workplace. Hence, it is the complete or comprehensive smoke-free policy (without any exceptions [designated smoking rooms] such as at airports, restaurants or hotels) that need to be enforced (not partial). There could be enhanced monitoring of implementation of such comprehensive smoke-free policies and fines/penalties should be substantially increased to discourage smoking in public places (and protect the non-smokers from SHS).

Reviewer #2: This revised paper still needs a substantial revision and editing.

1. There are published papers which used the same GATS-India-II data to examine the prevalence and correlates of SHS exposure in India (Tripathy 20201, 2020b). There is a need to discuss how this study differs from the published studies.

a. I found some question about sample size. The study by Tripathy (2020) used all observations in the GATS-India-II (N=74,037) and identified 64,538 non-smokers and 9,499 current smokers (including 7,647 daily smokers and 1,852 less-than-daily smokers). After deleting missing values, this study included 71,046 for the analysis of smoke-free policies at home. However, there are 11,321 current smokers according to Table 1. Why the sample size of current smokers is larger than the sample size (11,321 vs. 9,499) in the study by Tripathy (2020) even though the total sample size is smaller (71,046 vs. 74,037)? I suspect that this study might use a different definition of current smokers.

b. I also found some inconsistent findings. Tripathy (2020) reported that exposure to SHS in the home was 29.2% among all respondents and it was higher among females (30.4%) compared to males (28.1%); and that exposure to SHS in workplaces was 30% overall and it was higher among males (32.5%) compared to females (17.8%). However, this study yielded opposite results compared with Tripathy (2020).

Tripathy JP. Secondhand smoke exposure at home and public places among smokers and non-smokers in India: findings from the Global Adult Tobacco Survey 2016-17. Environ Sci Pollut Res Int. 2020a;27(6):6033-6041. doi: 10.1007/s11356-019-07341-x.

Tripathy JP. Smoke-free workplaces are associated with smoke-free homes in India: evidence for action. Environ Sci Pollut Res Int. 2020b;27(33):41405-41414. doi: 10.1007/s11356-020-10107-5.

2. (Page 2, abstract) The main aim of this study is to determine the correlates of smoke-free policies in the home and in the workplace. Before addressing the correlates, it is necessary to examine the prevalence of smoke-free policies in the home and in the workplace.

3. The second aim of this study is to assess the relationship between smoke-free policies and secondhand smoke (SHS) exposure. However, there is a lack of explanation for why examining this aim is important. Also, while the main aim of this study was analyzed using persons as the unit of the analysis, this aim was analyzed using states (30 states and 2 Union Territories) as the unit of the analysis. Moreover, while the main aim of this study was analyzed using the multivariable logistic regression model, this aim was analyzed using a simple Pearson’s correlation without adjusting for other confounding factors. If the authors really want to include this second aim in this paper, this aim should be examined in a much more rigorous way.

a. First, the definition of the SHS exposure variables, including what GATS questions were used to measure SHS exposure, should be explained clearly in the Methods section.

b. Second, a multivariable regression model which controls for other confounding factor should be used to analyze the association between smoke-free policies and SHS exposure.

c. Third, justification is needed for why the analysis needs to be conducted at state-level rather than at person-level.

4. (Page 2, abstract). More specific results about the correlates of smoke-free policies need to be provide in the Results section. For example, what are the prevalence estimates for smoke-free policies among all people aged 15+ in India? How does the prevalence differ by sub-groups stratified by each correlate? What are the key findings concerning the adjusted odds ratios for the associations between correlates and smoke-free policies?

5. (Page 2, abstract). The Conclusion section needs to be re-written to highlight the key findings.

6. Page 3, 1st paragraph: “GATS-2 also reports that 38.7% … of exposure to secondhand smoke (SHS) takes place at home, 30.2% … at workplace and 23% … occurs at any public place”. Before this sentence, it is important to let the readers know what was the prevalence of secondhand smoke exposure in India according to the GATS-2 data. Also, does this sentence mean that among people who were exposed to secondhand smoke, 38.7% were exposed at home, 30.2% were exposed at workplace, and 23% were exposed at public place. These 3 percentages only add up to 91.9%. What is the source of secondhand smoke exposure for the remaining 8.1%?

7. Page 3, 2nd paragraph “Although there is a progress of smoke-free policies … government still couldn’t reach its ultimate goal.” What is the ultimate goal of the National Tobacco Control Program launched in 2007-2008?

8. Page 3, 3rd paragraph.

a. Delete the wording “To our best knowledge”.

b. In the sentence “However, most of these previous studies have been focused on the prevalence and predictors of tobacco use [11, 18-20]“, the cited studies were not relevant because they are about secondhand smoke exposure rather than about tobacco use.

c. In the sentence “A study also observed that a smoke-free workplace was … associated with SHS exposure at home [21]“, Ref #19 can also be cited in addition to Ref #21.

d. In the sentence “A number of countries or community … the presence of policies that restrict smoking in the workplace and home may vary by … status [19, 22]“, Ref #19 should not be cited because it is about secondhand exposure rather than about smoke-free policies.

9. Page 4, Data paragraph, in the sentence “The analysis of smoking policies at workplace … but outside their home (adults)“, “smoking policies” should be corrected into ‘smoke-free policies”. Also, were “adults” defined as people aged 18+ or aged 15+?

10. Page 4, Section 2.3:

a. In the 1st sentence, delete “The status of” and “consisted of four categories”.

b. In the 2nd sentence, change the sentence “To determine the correlates of complete smoke-free policies at home and workplace classified into binary in the following ways “ to “Smoke-free policies at home and workplace were classified into binary in the following ways“.

c. If the outcome variable “smoke-free policies at home” is classified as a binary variable as stated in this section, why was this variable classified into a 4-group categorical variable in Table 1? The chi-square test for the association between 2-category smoke-free variable and the independent variable would have different results from the chi-square test for the association between 4-category smoke-free variable and the independent variable.

11. Page 5, 1st paragraph: The entire description of the independent variables was provided only in a supplementary table. Because the main aim of this study is to determine whether the selected independent variables are significant correlates of smoke-free policies, the independent variables should be explained in great detail in the Methods section including their names, definitions, and how they were measures. For example, for the smoking status variable, how were “smokers” defined? What GATS questions were used to define “smokers”? For the “knowledgeable of the children illness”, what GATS questions were used to define this variable?

12. Page 5, Section 2.4, the authors checked multicollinearity based on the values of SE for all predictors by citing Ref #23, which determined multicollinearity by the criteria of whether the magnitude of the SE for all predictors was > 0.5 by citing a study “Chan Y (2004) Biostatistics 202: logistic regression analysis. SingapMed J 45:149–153).” However, in the original work by Chan (2004), it was stated that measuring multicollinearity by inspecting the magnitude of the standard error (SE) of each variable is a simple but subjective technique because there is no fixed criterion on how small the SE should be but a matter of judgment. In the literature, a more widely used technique to diagnose multicollinearity is the variance inflation factor (VIF).

13. On page 6, 1st paragraph, the phrase “by conducting bivariate analysis“ and the sentence “A Chi-square test was used to assess the association … with selected factors. “ should be moved to the Methods section.

14. Page 6, 1st paragraph, in the sentence “… among the participants 54.4% had complete smoke-free policies at their home, 15.6% had partial … and 34% had no rules … (see Table 1)“, there are two problems. First, the sum of 54.4%, 15.6%, and 34% equals 104.0%, which is greater than 100%. There must be typos. Second, these numbers were not shown in Table 1. Similarly, in the next sentence, the reported percentages for smoke-free policy at workplace were not shown in Table 2.

15. Pages 6-7, the Results section needs to be substantially revised to improve the clarity. Some of the reported results (including Table 3) do not seem reasonable.

a. For example, in the sentence, “.. elderly people were more likely to enjoy a completely smoke-free environment at their home than the younger counterparts.”, who are the elderly people? Was the group aged 31-45 regarded as “elderly people”? What are the younger counterparts? Is this statement justified by the results in Table 3? I am puzzled why the AORs for the age groups 31-45, 46-60, and ≥60 were greater than 1 and highly significantly (p <.001) while the prevalence of smoke-free home rule was very similar across age groups (49.2% for the 15-30 group, 50.8% for the 31-45 group, 51.6% for the 46-60 group, and 50.6% for the ≥60 group. Was the sample weight used for the logistic regression analysis? I wonder whether the highly statistically significant results were due to the inflated sample size from using the sample weight. If this is the case, the models need to be re-estimated using the normalized weight.

b. A sentence states that “Female counterparts had less likely tendency to live in a complete smoke-free atmosphere in their residence compared to the male counterparts.” However, according to Table 1, the prevalence of smoke-free home rules was 50.7% for females and 50.0% for males. Their prevalence rates are quite close. It is peculiar why the AOR for female compared to male was less than 1 (AOR=0.569) and highly significant (p <.001).

c. Another sentence states that “Participants who lived in rural areas were 1.211 times more likely to report that they worked in a completely smoke-free environment than their counterparts.” However, according to Table 2, the prevalence of smoke-free workplace rule in rural areas was 52.1%, which is much lower than that in urban areas (69.4%). This illustrated that the reliability of Table 3 results is questionable.

d. The sentence “Participants who lived in the South, East, and West region of India had the higher tendency to report that they worked in a completely smoke-free environment compared to other regions” is not justifiable by the Table 3 results.

16. On page 7, the 2nd paragraph, the phrase “According to a research study,” should be deleted.

17. Pages 8-9, the Discussion section needs substantial revision by rigorously comparing the findings of this study with the literature, e.g,. how the findings are similar to or different from the previously published studies in the field, rather than repeating what has been presented in the Results section.

18. On page 8, the 2nd paragraph, the sentence “Married and separated people were more habituated to smoking in their workplace“ is inconsistent with the results shown in Table 3.

19. On page 9, the 1st paragraph, the meaning of the 4 sentences (“The research paper demonstrated that …. It could be detrimental for children in a two-fold way. They could fall victim …. But, if the children belonged to a highly educated and high income family, the exposure risk would decrease“) is not clear. How are these sentences relevant to the findings of this study?

20. On page 9, the 2nd paragraph, the sentence “For instance, … such as section 4 (smoking in public places-59.28%), 6-a (implementation level-68.57%) and 6-b (implementation level-52.85%) were observed to be violated to some extent [37].“ What does “implementation level” mean? How was it measured?

21. On Page 10, the authors cited a study conducted in the United States to explain the limitation of the GATS survey. This does not make sense.

22. Page 10, the last sentence — “… so that the projection of WHO about India being the only South Asian country to curb its smoking prevalence by 30% within 2025 will turn to be a success” is ambiguous. First, it needs to cite a reference about the WHO projected goal of smoking prevalence by 2014 for India. Second, it needs to explain what is the current smoking prevalence in India. This sentence seems to be contradictory to the information presented in the Introduction section (Page 3, lines 8-9): “Findings from the Global Adult Tobacco Survey 2 (GATS-2)-2017 indicate that 266.8 (28.6%) millions of all adults use tobacco in any form in India where 10.7% smoke tobacco”. Wasn’t India’s cigarette smoking prevalence rate in 2016-2017 10.7%?

23. Tables 1-2 need to add a row to show results for all the persons in the final study sample.

24. Tables 3-4 need to show the total sample size for the analysis.

25. In the Supplementary Table A1, for the age variable, “Up to 60 years” should be corrected into “More than 60 years”.

26. Inconsistent terms:

a. Smoke-free policies vs. smoking policies. The term “smoking policies” used in several places (e.g., page 4, line 2) should be corrected into “smoke-free policies”.

b. Outcome variables & independent variables (Section 2.3, line 1 and line 13) vs. dependent variables & explanatory variables (page 5, Section 2.4, line 3).

c. GATS2 vs. GATS 2

27. Inappropriate terms: “secondhand smoker’s exposure” should be corrected into “secondhand smoke exposure”.

7. PLOS authors have the option to publish the peer review history of their article (what does this mean?). If published, this will include your full peer review and any attached files.

Reviewer #1: **Yes: **Dr. Gaurang P. Nazar

Reviewer #2: No

---

## [Author Response · Author response to Decision Letter 1]

20 Aug 2021

Authors’ Response to Reviewers

Manuscript ID: PONE-D-20-41034R1

Title: Smoke-free policies in the home and in the workplace among Indian people: Evidence from Global Adult Tobacco Survey Data-2016/2017

Dear Respected Editor-in-Chief,

First of all, we thank the reviewer and the editor for their useful suggestions for improving the manuscript. We have now revised the manuscript accordingly. Thank you once again and looking forward to the final acceptance in your esteemed journal.

Reviewer (R1):

The authors have addressed a number of comments that were raised. The manuscript still requires significant copy-editing. There are some concerns regarding the terminologies used. The authors should ensure correct terms are used which are consistent with previous tobacco literature. There are some minor recommendations that I believe would improve the quality of the paper.

Abstract:

R1-1: Aim-edit “secondhand smoker’s exposure” to “secondhand smoke exposure”. Please ensure this changed is reflected throughout the manuscript. 

Authors: Thank you for your suggestions. We are already replaced the word “secondhand smoker’s exposure” to “secondhand smoke exposure” throughout the manuscript and update the revised manuscript. 

[Revised Manuscript Page #2; Aims (in Abstract)]

R1-2: Methods-edit “proportion of secondhand smoker’s exposure” to “proportion of participants exposed to secondhand smoke”. 

Authors: Thank you for your suggestions and giving us an opportunity to improve the revised manuscript. We are already replaced the word “proportion of secondhand smoker’s exposure” to “proportion of participants exposed to secondhand smoke” throughout the manuscript and update the revised manuscript.

[Revised Manuscript Page #2; Methods (in Abstract)]

R1-3: Results-edit “Results found that” to “We found that” edit “secondhand smoker’s exposure” to “secondhand smoke exposure”.

Authors: Thank you for your suggestions. We are already replaced the word “secondhand smoker’s exposure” to “secondhand smoke exposure” throughout the manuscript and update the revised manuscript. 

[Revised Manuscript Page #2; Results (in Abstract)]

R1-4: Conclusion-smoke free policies are already in place in India. Maybe authors could emphasize “comprehensive smoke free policies without any exceptions” and their “enhanced monitoring.” This could be a part of discussion and conclusion but authors should note - Penalty for violation of smoke-free legislation in public places is currently only INR 200 (approx. USD 2.7) which needs to be increased considerably taking into account inflation and rise in purchasing power/incomes. The proposed COTPA Amendment Bill (2020) proposes to increase this penalty to INR 2000 but this Bill has not yet been tabled in the Parliament.

Authors: Thank you for your suggestions. The manuscript has been revised and modified by considering your instruction. 

Introduction

R1-5: First para-6th line from bottom – remove “of them”

Authors: Thank you for your suggestions. The manuscript has been revised and modified by considering your instruction.

[Revised Manuscript Page #3; Section#1, First para-4-5th line from bottom]

R1-6: First para-5th line from bottom – “…tendency to smoke themselves but exposure to SHS…” can be replaced by “…tendency to smoke, exposure to SHS….”

Authors: Thank you for your suggestions. The manuscript has been revised and modified by considering your instruction. 

[Revised Manuscript Page #3; Section#1, First para-5th line from bottom]

R1-7: Second para – last sentence – “a progress of smoke-free policies” should be replaced by “progress in smoke-free policies”. “Government still couldn’t reach its ultimate goal” could be replaced by “the policies are still not comprehensive”

Authors: Thank you for your suggestions. The manuscript has been revised and modified by considering your instruction. 

[Revised Manuscript Page #3; Section#1, 2nd Para-Last sentence]

R1-8: Third para – “A number of countries or community-based Researches” could simply be replaced by “Previous research has”. And “secondhand smoker’s exposure” should be replaced at all places by “exposure to secondhand smoke”.

Authors: Thank you for your suggestions. The manuscript has been revised and modified by considering your instruction. 

[Revised Manuscript Page #3-4; Section#1, 3rd Para]

R1-9: Variables of the study – I think the household asset variable should be termed as the “wealth quintile” or “asset quintile”. This is typically the term used in similar research papers.

Authors: Thank you for your suggestions. We are already replaced the word “household asset” to “wealth quintile” throughout the manuscript and update the revised manuscript. 

[Revised Manuscript Page #5; Section#2, subsection#2.3, 2nd Para]

R1-10: Similarly, there is a problem with terminologies here “proportion of smoke-free policies and secondhand smoker’s exposure at both home and workplace”. This could be replaced by “proportion of participants living in smoke-free homes; working in smoke-free workplaces and proportion of participants exposed to secondhand smoke.” The authors should take care of these terminologies throughout the paper. The authors may also want to clarify here whether the proportion of participants exposed to SHS was restricted to non-smokers only.

Authors: Thank you for your suggestions. The manuscript has been revised and modified by considering your instruction. 

[Revised Manuscript Page #2; Section#2, subsection#2.3, last Para]

R1-11: The multicollinearity problem” could be replaced by just “multicollinearity”. The authors should run their regression models with ‘collin’ command (if using STATA) after running each logistic model and look for higher VIF values. While there is no rule of thumb for this, they could report that multicollinearity was assessed and all VIF values were less than 10, indicating no significant multicollinearity between explanatory variables. Multicollinearity between variables can lead to high SEs of coefficients and there are less chances of them being significant. Hence, only of such variables should be retained in the model. 

Apologies for not pointing this out earlier but in the statistical analysis section, the authors should report about using survey weights “svyset” to account for the complex multistaged GATS survey design.

Authors: Thank you for your suggestions. Yes, we are totally agreed with this reviewer. We are already run our logistic model with “collin” command (using STATA) after running logistic model and look VIF values for checking multicollinearity between independent variables. We observed that all VIF values were less than 10, indicating no significant multicollinearity between explanatory variables. Now, we have revised the Table and also updated the revised manuscript. Thanks again. 

Table 3. Multivariate logistic regression results

Variable Home Place Work Place

 p-value AOR (95% CI of AOR) VIF p-value AOR (95% CI of AOR) VIF

Tobacco Smoking Smoker® 1.05 1.11

status Non-smoker <0.001 3.285 (3.001,3.595) <0.001 1.627 (1.404, 1.886) 

Age group 15-30 - - 

 31-45 - - 

 46-60 - - 

 60+ - - 

Gender Male® - 1.11

 Female - 0.048 1.229 (1.002, 1.510) 

Marital status Single® 1.13 1.06

 Married 0.420 0.961 (0.874, 1.058) 0.604 0.954 (0.797, 1.141) 

 Separated/others 0.274 1.081 (0.940, 1.242) 0.934 0.984 (0.677, 1.431) 

Religion Hindhu® 1.01 1.01

 Muslim 0.074 0.885 (0.774, 1.012) 0.510 1.069 (0.876, 1.305) 

 Christian 0.901 0.987 (0.805, 1.211) 0.084 0.710 (0.482, 1.047) 

 Others <0.001 1.939 (1.650, 2.278) 0.589 1.099 (0.778, 1.554) 

Place of residence Urban® 1.21 1.19

 Rural <0.001 1.376 (1.253, 1.511) 0.015 1.245 (1.044, 1.484) 

Region North® 1.01 1.01

 Central 0.434 0.952 (0.840, 1.077) 0.170 0.866 (0.706, 1.064) 

 East <0.001 1.249 (1.107, 1.408) 0.003 1.424 (1.132, 1.791) 

 North-east 0.001 1.258 (1.095, 1.445) 0.162 1.181 (0.935, 1.491) 

 West <0.001 1.637 (1.423, 1.885) 0.001 1.671 (1.247, 2.239) 

 South <0.001 5.339 (4.764, 5.985) <0.001 1.836 (1.476, 2.285) 

Education Noeducation® 1.44 1.46

 Uptoprimary 0.004 1.127 (1.038, 1.222) 0.001 1.356 (1.124, 1.636) 

 Upto secondary <0.001 1.401 (1.286, 1.527) <0.001 1.831 (.501, 2.235) 

 Morethan <0.001 1.879 (1.695, 2.083) <0.001 2.317 (1.834, 2.928) 

 secondary 

Occupation Govt./Non-Govt. 1.07 1.10

 employee® 

 Dailywise/causal

laborer 0.008 0.853 (0.758, 0.959) <0.001 0.468 (0.381, 0.574) 

 Selfemployed 0.094 0.913 (0.821, 1.016) <0.001 0.559 (0.458, 0.684) 

 Homemaker 0.530 1.047 (0.908, 1.206) 0.001 2.064 (1.327, 3.208) 

 Student 0.001 0.821 (0.733, 0.919) 0.001 0.495 (0.329, 0.743) 

 Others 0.025 0.856 (0.746, 0.981) 0.095 0.629 (0.366, 1.083) 

Wealth quintile Q1(poorest)® 1.41 1.47

 Q2 0.623 0.977 (0.889, 1.072) 0.121 1.173 (0.959, 1.435) 

 Q3 0.170 1.076 (0.969, 1.195) 0.051 1.248 (0.998, 1.559) 

 Q4 0.018 1.176 (1.028, 1.346) 0.016 1.378 (1.061, 1.789) 

 Q5(richest) 0.108 1.103 (0.979, 1.243) 0.007 1.416 (1.101, 1.820) 

Otherpeople’s No® 1.02 

Smokingcauseserious

Illnessinchildren Yes 0.003 1.219 (1.068, 1.391) - 

- indicates that the variable is insignificant in bivariate analysis

[Revised Manuscript Page #15; Table #3]

In case of sampling weight, as per your suggestions, Sampling weights were applied and weighted estimates were calculated to account for the complex study design. Clustering and stratification was also accounted for by using svyset command in STATA. The following variables were used to apply clustering and stratification: gatscluster, gatsstrata and gatsweight. Now, we have revised and updated the statistical analysis section in the revised manuscript as follows: 

“Descriptive statistics was used to explore the scenario of the smoke-free policies in the home and in the workplace as well as the basic characteristics of the respondents. A Chi-square test was used to access the association between dependent and independent variables. Logistic regression analysis was used to determine the significant correlates of smoke-free policies at home and in the workplace and estimated adjusted odds ratio (AOR) along with 95% confidence interval of AOR. In this study, the multicollinearity among different independent variables was checked using variance inflation factor (VIF) values. If the all VIF values were less than 10, indicating there was no significant multicollinearity between independent variables [23]. Sampling weights were applied and weighted estimates were calculated to account for the complex study design. Clustering and stratification was also accounted for by using svyset command in STATA. The following variables were used to apply clustering and stratification: gatscluster, gatsstrata and gatsweight. Pearson correlation was also used to explore the relationship between smoke-free policies and the proportion of secondhand smoke exposure at both home and workplace.” 

[Revised Manuscript Page #5-6; section#2, subsection#2.4]

R1-12: In Results, Just want to point out here to the authors that SHS typically stands for “Secondhand Smoke” in tobacco literature and not “Secondhand Smoker’s”. This should be corrected throughout the paper and also in tables and their legends. The correct term should be “Secondhand smoke exposure” or SHS exposure.

Authors: Thank you for your suggestions and giving us an opportunity to improve the revised manuscript. We have already revised throughout the paper along with tables and their legends and updated the revised manuscript. Thanks again. 

R1-13: In Conclusion, this needs to be strengthened. As mentioned earlier, the authors studied complete smoke-free policy vs. no/partial smoke-free policy both at home and at workplace. Hence, it is the complete or comprehensive smoke-free policy (without any exceptions [designated smoking rooms] such as at airports, restaurants or hotels) that needs to be enforced (not partial). There could be enhanced monitoring of implementation of such comprehensive smoke-free policies and fines/penalties should be substantially increased to discourage smoking in public places (and protect the non-smokers from SHS).

Authors: Thank you for your suggestions. The manuscript has been revised and modified by considering your instruction. 

Reviewer (R2):

This revised paper still needs a substantial revision and editing.

R2-1: There are published papers which used the same GATS-India-II data to examine the prevalence and correlates of SHS exposure in India (Tripathy 20201, 2020b). There is a need to discuss how this study differs from the published studies. 

Authors: Thank you for asking this question and giving us an opportunity to improve the revised manuscript. The pattern of our current study is similar with Tripathy (20201, 2020b). But our objectives are totally differed with Tripathy (20201, 2020b). These are as follows: 

The main aim of this study was to determine the prevalence and correlates of smoke-free policies in the home and workplace. We have also investigated the state level relationship between smoke-free policies and secondhand smoke exposure. Whereas, Tripathy (2020) examined the relationship between smoke-free workplaces and smoke-free homes in India. He also examined the state level relationship between Smoke-free workplaces and smoke-free homes. Tripathy (2021) was examined the prevalence and correlates of SHS exposure at home and public place in India. 

R2-1a: I found some question about sample size. The study by Tripathy (2020) used all observations in the GATS-India-II (N=74,037) and identified 64,538 non-smokers and 9,499 current smokers (including 7,647 daily smokers and 1,852 less-than-daily smokers). After deleting missing values, this study included 71,046 for the analysis of smoke-free policies at home. However, there are 11,321 current smokers according to Table 1. Why the sample size of current smokers is larger than the sample size (11,321 vs. 9,499) in the study by Tripathy (2020) even though the total sample size is smaller (71,046 vs. 74,037)? I suspect that this study might use a different definition of current smokers.

Authors: Thank you for asking this question and giving us an opportunity to improve the revised manuscript. In order to classify the current tobacco smokers Tripathy (2020) considered the variable as B01: Do you currently smoke tobacco on a daily basis, less than daily, or not at all? Whereas, we have considered the variables as HH4E01: Does this person currently smoke tobacco, including bidis, cigarettes, hukkah, cigars, and pipes? For this reason, the numbers of current tobacco smokers are differed with Tripathy (2020). We have presented the frequency of both considering variables (B01 vs. HH4E01).

Before Cleaning (N=74,037) After Cleaning (N=71,046)

R2-1b: I also found some inconsistent findings. Tripathy (2020) reported that exposure to SHS in the home was 29.2% among all respondents and it was higher among females (30.4%) compared to males (28.1%); and that exposure to SHS in workplaces was 30% overall and it was higher among males (32.5%) compared to females (17.8%). However, this study yielded opposite results compared with Tripathy (2020).

Authors: Thank you for your suggestions. In our current study, we have presented the prevalence of smoke-free polices at home and workplace whereas, Tripathy (2020 Tripathy (2020) reported that exposure to SHS in the home and public place. We have carefully checked our results section; there is no any prevalence of exposure to SHS in the home and public place. We have only addressed some statistics from GATS 2 report in the introduction section as follows: 

“38.7% (38.1% male vs. 39.3% female) of exposure to secondhand smoke (SHS) takes place at home, 30.2% (32.7% male vs. 17.9% female) at workplace and 5.3% (8.1% male vs. 2.4% female) at government office, 5.6% (6.8% male vs. 4.4% female) at health care facilities, 7.4% (13.0% male vs. 1.6% female) at restaurants, and 13.3% (16.6% male vs. 9.9% female)”. 

Thanks again for your understanding. 

Reference

Tripathy JP. Secondhand smoke exposure at home and public places among smokers and non-smokers in India: findings from the Global Adult Tobacco Survey 2016-17. Environ Sci Pollut Res Int. 2020a; 27(6):6033-6041. doi: 10.1007/s11356-019-07341-x.

Tripathy JP. Smoke-free workplaces are associated with smoke-free homes in India: evidence for action. Environ Sci Pollut Res Int. 2020b;27(33):41405-41414. doi: 10.1007/s11356-020-10107-5.

R2-2: (Page 2, abstract) the main aim of this study is to determine the correlates of smoke-free policies in the home and in the workplace. Before addressing the correlates, it is necessary to examine the prevalence of smoke-free policies in the home and in the workplace.

Authors: Thank you for your suggestions and giving us an opportunity to improve the revised manuscript. Yes, we are totally agreed with this reviewer. Now, we have added a table as prevalence of prevalence of smoke-free policies in the home and in the workplace in the revised manuscript and updated the revised manuscript as follows: 

Table 1. Prevalence of smoke-free policies in the home and in the workplace

Characteristics Smoke-free policies in

 Home Workplace

 Yes, % No, % p-value Yes, % No, % p-value1

Overall 62.8 37.2 51.7 48.3 

Tobacco Smoking status <0.001 <0.001

Smoker 28.5 71.5 48.5 51.5 

Non-smoker 56.1 43.9 66.9 33.1 

Age group 0.944 0.676

15-30 51.8 48.2 63.6 36.4 

31-45 51.8 48.2 61.8 38.2 

46-60 51.3 48.7 62.7 37.3 

60+ 51.5 48.5 63.2 36.8 

Gender 0.205 <0.001

Male 52.2 47.8 61.6 38.4 

Female 51.1 48.9 68.9 31.1 

Religion <0.001 0.0086

Hindhu 51.9 48.1 63.5 36.5 

Muslim 45.4 54.6 57.1 42.9 

Christian 65.4 34.6 64.6 35.4 

Others 35.0 65.0 68.5 31.5 

Marital status <0.001 <0.001

Single 56.0 44.0 69.0 31.0 

Married 50.2 49.8 60.9 39.1 

Separated/others 52.6 47.4 58.9 41.1 

Place of residence <0.001 <0.001

Urban 46.1 53.9 56.0 44.0 

Rural 62.1 37.9 70.8 29.2 

Region <0.001 <0.001

North 56.3 53.7 62.3 37.7 

Central 38.3 61.7 52.7 47.3 

East 43.9 56.1 58.1 41.9 

North-east 43.3 56.7 55.9 44.1 

West 56.6 43.4 74.7 25.3 

South 77.9 22.1 72.2 27.8 

Education <0.001 <0.001

No education 41.5 58.5 41.5 58.5 

Up to primary 46.4 53.6 50.8 49.2 

Up to secondary 53.6 46.4 64.8 35.2 

More than secondary 65.1 34.9 76.3 23.7 

Occupation <0.001 <0.001

Govt./ Non-Govt. employee 60.7 39.3 75.5 24.5 

Daily wise/ causal labourer 46.8 53.2 46.6 53.4 

Self employed 49.1 50.9 58.2 41.8 

Homemaker 59.6 40.4 84.0 16.0 

Student 50.5 49.5 52.8 47.2 

Others (a) 50.1 49.9 58.0 42.0 

Household asset(b) <0.001 <0.001

Q1 (poorest) 39.1 60.9 46.1 53.9 

Q2 48.7 51.3 56.7 43.3 

Q3 54.5 45.5 62.1 37.9 

Q4 62.8 37.2 74.2 25.8 

Q5 (richest) 59.6 40.4 72.5 27.5 

Other people’s smoking cause serious illness in children <0.001 0.484

Yes 52.1 47.9 62.9 37.1 

No 43.4 56.6 60.0 40.0 

Q1: Poorest; Q2: Poorer; Q3: Middle; Q4: Richer; Q5: Richest.

[Revised Manuscript Page #1; Table#2]

R2-3: The second aim of this study is to assess the relationship between smoke-free policies and secondhand smoke (SHS) exposure. However, there is a lack of explanation for why examining this aim is important. Also, while the main aim of this study was analyzed using persons as the unit of the analysis, this aim was analyzed using states (30 states and 2 Union Territories) as the unit of the analysis. Moreover, while the main aim of this study was analyzed using the multivariable logistic regression model, this aim was analyzed using a simple Pearson’s correlation without adjusting for other confounding factors. If the authors really want to include this second aim in this paper, this aim should be examined in a much more rigorous way.

Authors: Thank you for your suggestions. We are totally agreed with you. In our current study, we do not adjust any confounding factor which was limitations of our study. We already address this issue in the limitations section in the revised manuscript. Thanks again.

[Revised Manuscript Page #9; Section#4, subsection#4.1]

R2-3a: First, the definition of the SHS exposure variables, including what GATS questions were used to measure SHS exposure, should be explained clearly in the Methods section.

Authors: Thank you for your suggestions. We are already add GATS questions to use to measure SHS exposure in the revised manuscript. 

[Revised Manuscript Page #5; Section#2, subsection#2.3]

R2-3b: Second, a multivariable regression model which controls for other confounding factor should be used to analyze the association between smoke-free policies and SHS exposure.

Authors: Thank you for your suggestions. We are totally agreed with you. In our current study, we do not adjust any confounding factor which was limitations of our study. We already address this issue in the limitations section in the revised manuscript. Thanks again. 

[Revised Manuscript Page #9; Section#4, subsection#4.1]

R2-3c: Third, justification is needed for why the analysis needs to be conducted at state-level rather than at person-level.

Authors: Thank you for your suggestions. The implementation of restrictions/rules about smoke-free policies varies state to state [Ali et al., 2020]. For this reason, we fixed our interest was to show the state relationship between the smoke-free policies and exposure to SHS at both home and workplace. Thanks again. 

R2-4: (Page 2, abstract). More specific results about the correlates of smoke-free policies need to be provide in the Results section. For example, what are the prevalence estimates for smoke-free policies among all people aged 15+ in India? How does the prevalence differ by sub-groups stratified by each correlate? What are the key findings concerning the adjusted odds ratios for the associations between correlates and smoke-free policies?

Authors: Thank you for your suggestions. We have already updated it in the revised manuscript. Thanks again. 

R2-5: (Page 2, abstract). The Conclusion section needs to be re-written to highlight the key findings.

Authors: Thank you for your suggestions. The manuscript has been revised and modified by considering your instruction. 

R2-6: Page 3, 1st paragraph: “GATS-2 also reports that 38.7% … of exposure to secondhand smoke (SHS) takes place at home, 30.2% … at workplace and 23% … occurs at any public place”. Before this sentence, it is important to let the readers know what was the prevalence of secondhand smoke exposure in India according to the GATS-2 data. Also, does this sentence mean that among people who were exposed to secondhand smoke, 38.7% were exposed at home, 30.2% were exposed at workplace, and 23% were exposed at public place. These 3 percentages only add up to 91.9%. What is the source of secondhand smoke exposure for the remaining 8.1%?

Authors: Thanks again for giving us your suggestions to improve the revised manuscript. We have already fixed the above sentence and updated the revised manuscript as follows: 

“According to the GATS 2 (2016-2017) data, 38.7% (38.1% male vs. 39.3% female) of exposure to secondhand smoke (SHS) takes place at home, 30.2% (32.7% male vs. 17.9% female) at workplace and 5.3% (8.1% male vs. 2.4% female) at government office, 5.6% (6.8% male vs. 4.4% female) at health care facilities, 7.4% (13.0% male vs. 1.6% female) at restaurants, and 13.3% (16.6% male vs. 9.9% female) [6].”

[Revised Manuscript Page #3; Section#1, 1st Para]

R2-7: Page 3, 2nd paragraph “Although there is a progress of smoke-free policies … government still couldn’t reach its ultimate goal.” What is the ultimate goal of the National Tobacco Control Program launched in 2007-2008?

Authors: Thank you for your suggestions. We have already updated the above sentence as per guidance of Reviewer#1. Thanks again. 

R2-8: Page 3, 3rd paragraph.

R2-8a: Delete the wording “To our best knowledge”.

Authors: Thank you for your suggestions. We have already removed the word “To our best knowledge” from our revised manuscript. Thanks again. 

R2-8b: In the sentence “However, most of these previous studies have been focused on the prevalence and predictors of tobacco use [11, 18-20]”, the cited studies were not relevant because they are about secondhand smoke exposure rather than about tobacco use.

Authors: Thank you for your suggestions and giving us an opportunity to improve the revised manuscript. Yes, we are totally agreed with this reviewer. Now, we have fixed it and updated the revised manuscript. Thanks again.

R2-8c: In the sentence “A study also observed that a smoke-free workplace was … associated with SHS exposure at home [21] “, Ref #19 can also be cited in addition to Ref #21.

Authors: Thank you for your suggestions. As your suggestions, we have already cited Ref#19 instead of Ref#21 and updated the revised manuscript. Thanks again. 

R2-8d: In the sentence “A number of countries or community … the presence of policies that restrict smoking in the workplace and home may vary by … status [19, 22]“, Ref #19 should not be cited because it is about secondhand exposure rather than about smoke-free policies.

Authors: Thank you for your suggestions. As your suggestions, we have already cited Ref#22 in the above mentioned sentence instead of Ref#19, 22 and updated the revised manuscript. Thanks again.

R2-9: Page 4, Data paragraph, in the sentence “The analysis of smoking policies at workplace … but outside their home (adults)“, “smoking policies” should be corrected into ‘smoke-free policies”. Also, were “adults” defined as people aged 18+ or aged 15+?

Authors: Thank you for your suggestions. We have already replaced the word “smoking policies” by “smoking-free policies” in the revised manuscript. In regarding adult, we are defined the word adult as people aged 15+. Thanks again. 

R2-10: Page 4, Section 2.3:

R2-10a: In the 1st sentence, delete “The status of” and “consisted of four categories”.

Authors: Thank you for your suggestions. We have already removed the word “The status of” and “consisted of four categories” from our revised manuscript. Thanks again. 

R2-10b: In the 2nd sentence, change the sentence “To determine the correlates of complete smoke-free policies at home and workplace classified into binary in the following ways “ to “Smoke-free policies at home and workplace were classified into binary in the following ways”.

Authors: Thank you for your suggestions. We have already fixed the above sentence and updated the revised manuscript. Thanks again. 

[Revised Manuscript Page #4; Section#2, Subsection#2.3, 2nd sentence]

R2-10c. If the outcome variable “smoke-free policies at home” is classified as a binary variable as stated in this section, why was this variable classified into a 4-group categorical variable in Table 1? The chi-square test for the association between 2-category smoke-free variable and the independent variable would have different results from the chi-square test for the association between 4-category smoke-free variable and the independent variable.

Authors: Thank you for your suggestions and giving us an opportunity to improve the revised manuscript. We are totally agreed with this reviewer. Now, we have made a Table 2 with the combination of previous Table 1 and Table 2 for two categories of outcome instead of 4-categories and updated the revised manuscript. Thanks again. 

Table 1. Prevalence of smoke-free policies in the home and in the workplace

Characteristics Smoke-free policies in

 Home Workplace

 Yes, % No, % p-value Yes, % No, % p-value1

Overall 62.8 37.2 51.7 48.3 

Tobacco Smoking status <0.001 <0.001

Smoker 28.5 71.5 48.5 51.5 

Non-smoker 56.1 43.9 66.9 33.1 

Age group 0.944 0.676

15-30 51.8 48.2 63.6 36.4 

31-45 51.8 48.2 61.8 38.2 

46-60 51.3 48.7 62.7 37.3 

60+ 51.5 48.5 63.2 36.8 

Gender 0.205 <0.001

Male 52.2 47.8 61.6 38.4 

Female 51.1 48.9 68.9 31.1 

Religion <0.001 0.0086

Hindhu 51.9 48.1 63.5 36.5 

Muslim 45.4 54.6 57.1 42.9 

Christian 65.4 34.6 64.6 35.4 

Others 35.0 65.0 68.5 31.5 

Marital status <0.001 <0.001

Single 56.0 44.0 69.0 31.0 

Married 50.2 49.8 60.9 39.1 

Separated/others 52.6 47.4 58.9 41.1 

Place of residence <0.001 <0.001

Urban 46.1 53.9 56.0 44.0 

Rural 62.1 37.9 70.8 29.2 

Region <0.001 <0.001

North 56.3 53.7 62.3 37.7 

Central 38.3 61.7 52.7 47.3 

East 43.9 56.1 58.1 41.9 

North-east 43.3 56.7 55.9 44.1 

West 56.6 43.4 74.7 25.3 

South 77.9 22.1 72.2 27.8 

Education <0.001 <0.001

No education 41.5 58.5 41.5 58.5 

Up to primary 46.4 53.6 50.8 49.2 

Up to secondary 53.6 46.4 64.8 35.2 

More than secondary 65.1 34.9 76.3 23.7 

Occupation <0.001 <0.001

Govt./ Non-Govt. employee 60.7 39.3 75.5 24.5 

Daily wise/ causal labourer 46.8 53.2 46.6 53.4 

Self employed 49.1 50.9 58.2 41.8 

Homemaker 59.6 40.4 84.0 16.0 

Student 50.5 49.5 52.8 47.2 

Others (a) 50.1 49.9 58.0 42.0 

Household asset(b) <0.001 <0.001

Q1 (poorest) 39.1 60.9 46.1 53.9 

Q2 48.7 51.3 56.7 43.3 

Q3 54.5 45.5 62.1 37.9 

Q4 62.8 37.2 74.2 25.8 

Q5 (richest) 59.6 40.4 72.5 27.5 

Other people’s smoking cause serious illness in children <0.001 0.484

Yes 52.1 47.9 62.9 37.1 

No 43.4 56.6 60.0 40.0 

Q1: Poorest; Q2: Poorer; Q3: Middle; Q4: Richer; Q5: Richest.

[Revised Manuscript Page #15; Table#2]

R2-11: Page 5, 1st paragraph: The entire description of the independent variables was provided only in a supplementary table. Because the main aim of this study is to determine whether the selected independent variables are significant correlates of smoke-free policies, the independent variables should be explained in great detail in the Methods section including their names, definitions, and how they were measures. For example, for the smoking status variable, how were “smokers” defined? What GATS questions were used to define “smokers”? For the “knowledgeable of the children illness”, what GATS questions were used to define this variable?

Authors: Thank you for your suggestions and giving us an opportunity to improve the revised manuscript. We are totally agreed with this reviewer. Now, we have added a Table in the method section for description of independent variables and update the revised manuscript. Thanks again. 

[Revised Manuscript Page #5; Table#1]

R2-12: Page 5, Section 2.4, the authors checked multi-collinearity based on the values of SE for all predictors by citing Ref #23, which determined multi-collinearity by the criteria of whether the magnitude of the SE for all predictors was > 0.5 by citing a study “Chan Y (2004) Biostatistics 202: logistic regression analysis. SingapMed J 45:149–153).” However, in the original work by Chan (2004), it was stated that measuring multicollinearity by inspecting the magnitude of the standard error (SE) of each variable is a simple but subjective technique because there is no fixed criterion on how small the SE should be but a matter of judgment. In the literature, a more widely used technique to diagnose multicollinearity is the variance inflation factor (VIF).

Authors: Thank you for your suggestions. Yes, we are totally agreed with this reviewer. We are already run our logistic model with “collin” command (using STATA) after running logistic model and look VIF values for checking multicollinearity between independent variables. We observed that all VIF values were less than 10, indicating no significant multicollinearity between explanatory variables. Now, we have revised the Table and also updated the revised manuscript. Thanks again. 

Table 3. Multivariate logistic regression results

Variable Home Place Work Place

 p-value AOR (95% CI of AOR) VIF p-value AOR (95% CI of AOR) VIF

Tobacco Smoking Smoker® 1.05 1.11

status Non-smoker <0.001 3.285 (3.001,3.595) <0.001 1.627 (1.404, 1.886) 

Age group 15-30 - - 

 31-45 - - 

 46-60 - - 

 60+ - - 

Gender Male® - 1.11

 Female - 0.048 1.229 (1.002, 1.510) 

Marital status Single® 1.13 1.06

 Married 0.420 0.961 (0.874, 1.058) 0.604 0.954 (0.797, 1.141) 

 Separated/others 0.274 1.081 (0.940, 1.242) 0.934 0.984 (0.677, 1.431) 

Religion Hindhu® 1.01 1.01

 Muslim 0.074 0.885 (0.774, 1.012) 0.510 1.069 (0.876, 1.305) 

 Christian 0.901 0.987 (0.805, 1.211) 0.084 0.710 (0.482, 1.047) 

 Others <0.001 1.939 (1.650, 2.278) 0.589 1.099 (0.778, 1.554) 

Place of residence Urban® 1.21 1.19

 Rural <0.001 1.376 (1.253, 1.511) 0.015 1.245 (1.044, 1.484) 

Region North® 1.01 1.01

 Central 0.434 0.952 (0.840, 1.077) 0.170 0.866 (0.706, 1.064) 

 East <0.001 1.249 (1.107, 1.408) 0.003 1.424 (1.132, 1.791) 

 North-east 0.001 1.258 (1.095, 1.445) 0.162 1.181 (0.935, 1.491) 

 West <0.001 1.637 (1.423, 1.885) 0.001 1.671 (1.247, 2.239) 

 South <0.001 5.339 (4.764, 5.985) <0.001 1.836 (1.476, 2.285) 

Education Noeducation® 1.44 1.46

 Uptoprimary 0.004 1.127 (1.038, 1.222) 0.001 1.356 (1.124, 1.636) 

 Upto secondary <0.001 1.401 (1.286, 1.527) <0.001 1.831 (.501, 2.235) 

 Morethan <0.001 1.879 (1.695, 2.083) <0.001 2.317 (1.834, 2.928) 

 secondary 

Occupation Govt./Non-Govt. 1.07 1.10

 employee® 

 Dailywise/causal

laborer 0.008 0.853 (0.758, 0.959) <0.001 0.468 (0.381, 0.574) 

 Selfemployed 0.094 0.913 (0.821, 1.016) <0.001 0.559 (0.458, 0.684) 

 Homemaker 0.530 1.047 (0.908, 1.206) 0.001 2.064 (1.327, 3.208) 

 Student 0.001 0.821 (0.733, 0.919) 0.001 0.495 (0.329, 0.743) 

 Others 0.025 0.856 (0.746, 0.981) 0.095 0.629 (0.366, 1.083) 

Wealth quintile Q1(poorest)® 1.41 1.47

 Q2 0.623 0.977 (0.889, 1.072) 0.121 1.173 (0.959, 1.435) 

 Q3 0.170 1.076 (0.969, 1.195) 0.051 1.248 (0.998, 1.559) 

 Q4 0.018 1.176 (1.028, 1.346) 0.016 1.378 (1.061, 1.789) 

 Q5(richest) 0.108 1.103 (0.979, 1.243) 0.007 1.416 (1.101, 1.820) 

Otherpeople’s No® 1.02 

Smokingcauseserious

Illnessinchildren Yes 0.003 1.219 (1.068, 1.391) - 

- indicates that the variable is insignificant in bivariate analysis

[Revised Manuscript Page #15; Table #2]

R2-13: On page 6, 1st paragraph, the phrase “by conducting bivariate analysis“ and the sentence “A Chi-square test was used to assess the association … with selected factors. “ should be moved to the Methods section.

Authors: Thank you for your suggestions. We are already moved this sentence from Results section to Methods section and updated the revised manuscript.

[Revised Manuscript Page #5; Section#2, Subsection#2.4]

R2-14: Page 6, 1st paragraph, in the sentence “… among the participants 54.4% had complete smoke-free policies at their home, 15.6% had partial … and 34% had no rules … (see Table 1)“, there are two problems. First, the sum of 54.4%, 15.6%, and 34% equals 104.0%, which is greater than 100%. There must be typos. Second, these numbers were not shown in Table 1. Similarly, in the next sentence, the reported percentages for smoke-free policy at workplace were not shown in Table 2.

Authors: Thank you for your suggestions. The manuscript has been revised and modified by considering your instruction. Now, we have made a Table 2 with the combination of previous Table 1 and Table 2 for two categories of outcome instead of 4-categories and updated the revised manuscript. Thanks again.

Table 2. Prevalence of smoke-free policies in the home and in the workplace

Characteristics Smoke-free policies in

 Home Workplace

 Yes, % No, % p-value Yes, % No, % p-value1

Overall 62.8 37.2 51.7 48.3 

Tobacco Smoking status <0.001 <0.001

Smoker 28.5 71.5 48.5 51.5 

Non-smoker 56.1 43.9 66.9 33.1 

Age group 0.944 0.676

15-30 51.8 48.2 63.6 36.4 

31-45 51.8 48.2 61.8 38.2 

46-60 51.3 48.7 62.7 37.3 

60+ 51.5 48.5 63.2 36.8 

Gender 0.205 <0.001

Male 52.2 47.8 61.6 38.4 

Female 51.1 48.9 68.9 31.1 

Religion <0.001 0.0086

Hindhu 51.9 48.1 63.5 36.5 

Muslim 45.4 54.6 57.1 42.9 

Christian 65.4 34.6 64.6 35.4 

Others 35.0 65.0 68.5 31.5 

Marital status <0.001 <0.001

Single 56.0 44.0 69.0 31.0 

Married 50.2 49.8 60.9 39.1 

Separated/others 52.6 47.4 58.9 41.1 

Place of residence <0.001 <0.001

Urban 46.1 53.9 56.0 44.0 

Rural 62.1 37.9 70.8 29.2 

Region <0.001 <0.001

North 56.3 53.7 62.3 37.7 

Central 38.3 61.7 52.7 47.3 

East 43.9 56.1 58.1 41.9 

North-east 43.3 56.7 55.9 44.1 

West 56.6 43.4 74.7 25.3 

South 77.9 22.1 72.2 27.8 

Education <0.001 <0.001

No education 41.5 58.5 41.5 58.5 

Up to primary 46.4 53.6 50.8 49.2 

Up to secondary 53.6 46.4 64.8 35.2 

More than secondary 65.1 34.9 76.3 23.7 

Occupation <0.001 <0.001

Govt./ Non-Govt. employee 60.7 39.3 75.5 24.5 

Daily wise/ causal labourer 46.8 53.2 46.6 53.4 

Self employed 49.1 50.9 58.2 41.8 

Homemaker 59.6 40.4 84.0 16.0 

Student 50.5 49.5 52.8 47.2 

Others (a) 50.1 49.9 58.0 42.0 

Household asset(b) <0.001 <0.001

Q1 (poorest) 39.1 60.9 46.1 53.9 

Q2 48.7 51.3 56.7 43.3 

Q3 54.5 45.5 62.1 37.9 

Q4 62.8 37.2 74.2 25.8 

Q5 (richest) 59.6 40.4 72.5 27.5 

Other people’s smoking cause serious illness in children <0.001 0.484

Yes 52.1 47.9 62.9 37.1 

No 43.4 56.6 60.0 40.0 

Q1: Poorest; Q2: Poorer; Q3: Middle; Q4: Richer; Q5: Richest.

[Revised Manuscript Page #15; Table#2]

R2-15: Pages 6-7, the Results section needs to be substantially revised to improve the clarity. Some of the reported results (including Table 3) do not seem reasonable.

a. For example, in the sentence, “.. elderly people were more likely to enjoy a completely smoke-free environment at their home than the younger counterparts.”, who are the elderly people? Was the group aged 31-45 regarded as “elderly people”? What are the younger counterparts? Is this statement justified by the results in Table 3? I am puzzled why the AORs for the age groups 31-45, 46-60, and ≥60 were greater than 1 and highly significantly (p <.001) while the prevalence of smoke-free home rule was very similar across age groups (49.2% for the 15-30 group, 50.8% for the 31-45 group, 51.6% for the 46-60 group, and 50.6% for the ≥60 group. Was the sample weight used for the logistic regression analysis? I wonder whether the highly statistically significant results were due to the inflated sample size from using the sample weight. If this is the case, the models need to be re-estimated using the normalized weight.

b. A sentence states that “Female counterparts had less likely tendency to live in a complete smoke-free atmosphere in their residence compared to the male counterparts.” However, according to Table 1, the prevalence of smoke-free home rules was 50.7% for females and 50.0% for males. Their prevalence rates are quite close. It is peculiar why the AOR for female compared to male was less than 1 (AOR=0.569) and highly significant (p <.001).

c. Another sentence states that “Participants who lived in rural areas were 1.211 times more likely to report that they worked in a completely smoke-free environment than their counterparts.” However, according to Table 2, the prevalence of smoke-free workplace rule in rural areas was 52.1%, which is much lower than that in urban areas (69.4%). This illustrated that the reliability of Table 3 results is questionable.

d. The sentence “Participants who lived in the South, East, and West region of India had the higher tendency to report that they worked in a completely smoke-free environment compared to other regions” is not justifiable by the Table 3 results.

Authors: Thank you for your suggestions and giving us an opportunity to improve the revised manuscript. We have already revised the results section and update the revised manuscript. Thanks again. 

[Revised Manuscript Page #6-7; Section#3 (Results)]

R2-16: On page 7, the 2nd paragraph, the phrase “According to a research study,” should be deleted.

Authors: Thank you for your suggestions. The manuscript has been revised and modified by considering your instruction. 

R2-17: Pages 8-9, the Discussion section needs substantial revision by rigorously comparing the findings of this study with the literature, e.g,. how the findings are similar to or different from the previously published studies in the field, rather than repeating what has been presented in the Results section.

Authors: Thank you for your suggestions. The manuscript has been revised and modified by considering your instruction. 

R2-18: On page 8, the 2nd paragraph, the sentence “Married and separated people were more habituated to smoking in their workplace” is inconsistent with the results shown in Table 3.

Authors: Thank you for your suggestions. We have already revised this sentence in discussion section and updated the revised manuscript. Thanks again. 

R2-19: On page 9, the 1st paragraph, the meaning of the 4 sentences (“The research paper demonstrated that …. It could be detrimental for children in a two-fold way. They could fall victim …. But, if the children belonged to a highly educated and high income family, the exposure risk would decrease“) is not clear. How are these sentences relevant to the findings of this study?

Authors: Thank you for your suggestions. We are totally agreed with this reviewer. Now, we have removed these sentences from discussion and also revised the discussion section in the revised manuscript. Thanks again. 

R2-20: On page 9, the 2nd paragraph, the sentence “For instance, such as section 4 (smoking in public places-59.28%), 6-a (implementation level-68.57%) and 6-b (implementation level-52.85%) were observed to be violated to some extent [37].“ What does “implementation level” mean? How was it measured?

Authors: Thank you for your suggestions. We have collected the above statistics from Ali et al. (2020). They used an assessment tools to measure the implementation tools which were explained in details in the following paper: 

Ali, I., Patthi, B., Singla, A., Dhama, K., Muchhal, M., Rajeev, A., ...& Khan, A. (2020). Assessment of implementation and compliance of (COTPA) Cigarette and Other Tobacco Products Act (2003) in open places of Delhi. Journal of Family Medicine and Primary Care, 9(6), 3094.

R2-21: On Page 10, the authors cited a study conducted in the United States to explain the limitation of the GATS survey. This does not make sense. 

Authors: Thank you for your suggestions. We are extremely sorry for this great mistake. Now, we have removed this sentence from strengths and limitations and updated the revised manuscript. Thanks again. 

R2-22: Page 10, the last sentence — “… so that the projection of WHO about India being the only South Asian country to curb its smoking prevalence by 30% within 2025 will turn to be a success” is ambiguous. First, it needs to cite a reference about the WHO projected goal of smoking prevalence by 2014 for India. Second, it needs to explain what the current smoking prevalence in India is. This sentence seems to be contradictory to the information presented in the Introduction section (Page 3, lines 8-9): “Findings from the Global Adult Tobacco Survey 2 (GATS-2)-2017 indicate that 266.8 (28.6%) millions of all adults use tobacco in any form in India where 10.7% smoke tobacco”. Wasn’t India’s cigarette smoking prevalence rate in 2016-2017 10.7%?

Authors: Thank you for your suggestions. We are totally agreed this reviewer. We have omitted this sentence from conclusion and updated the revised manuscript. Thanks again. 

R2-23: Tables 1-2 need to add a row to show results for all the persons in the final study sample.

Authors: Thank you for your suggestions. We have already made a new Table with the combination of Table 1 and Table 2 and add a row to show results for all the persons in the final study sample as follows: 

Characteristics Living in a complete smoke-free policies (n= 15,254) Working in a complete smoke-free policies (n= 71,046)

 Yes % No, % p-value Yes, % No, % p-value1

Overall 62.8 37.2 51.7 48.3 

R2-24: Tables 3-4 need to show the total sample size for the analysis.

Authors: Thank you for your suggestions. The manuscript has been revised and modified by considering your instruction.

R2-25: In the Supplementary Table A1, for the age variable, “Up to 60 years” should be corrected into “More than 60 years”.

Authors: Thank you for your suggestions. We have already replaced Up to 60 years by More than 60 years in the In the Supplementary Table A1 and updated the revised manuscript. Thanks again. 

R2-26: Inconsistent terms:

R2-26a: Smoke-free policies vs. smoking policies. The term “smoking policies” used in several places (e.g., page 4, line 2) should be corrected into “smoke-free policies”.

Authors: Thank you for your suggestions. We have already replaced the word “smoking policies” by smoke-free policies throughout the manuscript and updated the revised manuscript. Thanks again. 

[Revised Manuscript Page #4; Section#2, Subsection#2.1]

R2-26b: Outcome variables & independent variables (Section 2.3, line 1 and line 13) vs. dependent variables & explanatory variables (page 5, Section 2.4, line 3).

Authors: Thank you for your suggestions. We have already fixed it and updated the revised manuscript. Thanks again.

[Revised Manuscript Page #4, 5; Section#2, Subsection#2.3, 2.4]

R2-26c: GATS2 vs. GATS 2

Authors: Thank you for your suggestions. We have already replaced the word “GATS2” by “GATS 2” throughout the manuscript. Thanks again. 

R2-27: Inappropriate terms: “secondhand smoker’s exposure” should be corrected into “secondhand smoke exposure”.

Authors: Thank you for your suggestions. The manuscript has been revised and modified by considering your instruction.

---

## [Decision Letter · Decision Letter 2]

11 Apr 2022

PONE-D-20-41034R2Smoke-free policies in the home and in the workplace among Indian people: Evidence from Global Adult Tobacco Survey Data-2016/2017PLOS ONE

Dear Dr. Alauddin,

Thank you for submitting your manuscript to PLOS ONE. After careful consideration, we feel that it has merit but does not fully meet PLOS ONE’s publication criteria as it currently stands. Therefore, we invite you to submit a revised version of the manuscript that addresses the points raised during the review process.

Two new reviewers were invited to assess the revised  manuscript as the previous reviewers were unavailable. The reviewers believe that while the authors have addressed the comments of the previous reviewer, some additional changes are required to further improved the manuscript.

Please note that the reviewers believe that overall quality of language in the mansucript can be improved. PLOS ONE cannot provide copy editing for accepted mansucript and therefore, we suggest you thoroughly copyedit your manuscript for language usage, spelling, and grammar. If you do not know anyone who can help you do this, you may wish to consider employing a professional scientific editing service.  

We look forward to receiving your revised manuscript.

Kind regards,

Lucinda Shen

Staff Editor

PLOS ONE

Journal Requirements:

Reviewers' comments:

Reviewer's Responses to Questions

**Comments to the Author**

1. If the authors have adequately addressed your comments raised in a previous round of review and you feel that this manuscript is now acceptable for publication, you may indicate that here to bypass the “Comments to the Author” section, enter your conflict of interest statement in the “Confidential to Editor” section, and submit your "Accept" recommendation.

Reviewer #3: (No Response)

Reviewer #4: All comments have been addressed

2. Is the manuscript technically sound, and do the data support the conclusions?

Reviewer #3: Yes

Reviewer #4: Yes

3. Has the statistical analysis been performed appropriately and rigorously? 

Reviewer #3: No

Reviewer #4: Yes

4. Have the authors made all data underlying the findings in their manuscript fully available?

Reviewer #3: Yes

Reviewer #4: Yes

5. Is the manuscript presented in an intelligible fashion and written in standard English?

Reviewer #3: Yes

Reviewer #4: No

6. Review Comments to the Author

Reviewer #3: - More should be said about the smoke-free policies and their enforcement. It is not clear why, if there are smoke-free policies, people still smoke at workplaces, e.g. More should be said about how policies have been implemented, i.e., if there are fines in case of non-compliance, and it there is some monitoring.

- The dependent variable should not be called smoke-free “policies” since authors are not evaluating policies. Instead, it should be named smoking “norms”, e.g.

- Authors have been including several variables, but do not provide any justification to include them. It seems more they have been “fishing” factors, without any clear theoretical background to consider specific variables. In the same line, the Discussion fails to provide explanations for why women, richer and more educated people live and work in more smoke-free environments.

- The correlation between SHS and smoke-free policies is not convincing as it does not consider potential confounders, as this is the case in the main analysis.

- There is a clear problem of reverse causation when using the smoker variable, as living in a more smoke-free environment reduces the likelihood to smoke. To avoid this problem, authors may use the prevalence of smoking in the region or municipality, e.g.

- More generally, the implementation of norms is related to contextual variables, namely, the degree of enforcement of policies, cultural perceptions about smoking, other anti-tobacco policies (taxes, sales bans to minors, etc), or communication strategies by local authorities.

- The relevance of the paper is not clearly highlighted. Authors should state why measuring the determinants of smoke-free environments is relevant from a public health viewpoint, and how results may be relevant for decision making. We miss a section with the policy implications of the paper.

Reviewer #4: General comments

The authors have responded to all comments by the previous reviewers. However, some major language revision are still needed (especially the results and discussion sections) to help improve the readability of the manuscript.

Abstract

Some of the sentences needs restructuring to make it read better.

Introduction

As mentioned by previous reviewers, please elaborate on the importance of the second objective (i.e. correlation between state-level smoke free policy and SHS exposure).

Materials and Methods

Results

In many places, the sentences need to be rewritten. For example: Page 6 Line 15 “ About 28.5% of tobacco smokers were kept to complete smoke-free policies in the home, whereas 48.5% of smokers in the workplace”. The meaning of this sentence is unclear. Do the authors mean “25% and 48.5% of tobacco smokers have smoke-free policies in their homes and workplace respectively?”

Table 2:

To indicate which variables were adjusted for in the multivariable model?

I suggest that the table be titled differently to better describe its contents. A suggestion is “Correlates of smoke-free policies in the home and the workplace in India”

Th occupation ‘daily wise/causal labourer’ is probably meant to be ‘daily wage/casual labourer’?

Table 3:

As pointed out by a previous reviewer, the word multivariable is appropriate here rather than multivariate logistic regression (The terms are not interchangeable).

ORs and the 95% CIs need be expressed up to 2 decimals only.

The use of the ‘’registered’ ® symbol to indicate the reference category is rather inappropriate (Or does this symbol appear only on my computer?). The authors may put ‘1’, ‘Reference’ or ‘.ref’ in the AOR for the reference category instead to indicate this.

Table 4:

The table should be given a more descriptive title e.g. “Correlations between smoke-free policies in the home and workplace and secondhand smoke exposure”

Only half of the table cells need to be filled as the other half is redundant (have the same values).

The correlation coefficient may be reported to only 2 decimals.

‘SHS at home’ should be ‘SHS exposure at home’ (similarly for workplace and public places).

In the footnote, SHS should stand for secondhand smoke instead of secondhand smoker.

Discussion

Suggest to first summarize the finding and then interpretation and discussion in each paragraph. For example on Pg 8 para 3, it would help to state the correlation finding first before explaining the possible reasons why it is so.

The language style throughout the section need to be given a another round of proofreading.

- Example on Pg 8 “Literature suggests the fact that” should be “literature suggest that”

Pg 8, “Other ocupations tend to..” does “other occupations” mean non-professional occupations?

Pg 8: The statement “unspoiled , contaminated environment”, unspoiled and contaminated are contradictory, please check this statement.

Conclusion

The implications of the study findings with regards to tobacco control policy as commented by the previous reviewer are still not spelled out. What can the authorities do with the findings? Make more specific suggestions/recommendations.

7. PLOS authors have the option to publish the peer review history of their article (what does this mean?). If published, this will include your full peer review and any attached files.

Reviewer #3: **Yes: **Julian Perelman

Reviewer #4: No

---

## [Author Response · Author response to Decision Letter 2]

5 Jun 2022

Authors’ Response to Reviewers

Manuscript ID: PONE-D-20-41034R2

Title: Smoke-free policies in the home and in the workplace among Indian people: Evidence from Global Adult Tobacco Survey Data-2017

Dear Respected Editor-in-Chief,

First of all, we thank the reviewer and the editor for their useful suggestions for improving the manuscript. We have now revised the manuscript accordingly. Thank you once again and looking forward to the final acceptance in your esteemed journal.

Reviewer (R3):

More should be said about the smoke-free policies and their enforcement. It is not clear why, if there are smoke-free policies, people still smoke at workplaces, e.g. More should be said about how policies have been implemented, i.e., if there are fines in case of non-compliance, and it there is some monitoring. I have added some lines. Please Check:

R3-1: The dependent variable should not be called smoke-free “policies” since authors are not evaluating policies. Instead, it should be named smoking “norms”, e.g.

Authors: Thank you very much for your valuable suggestions and for giving us an opportunity to improve the revised manuscript. the comment. As per your suggestions, we have replaced the word “smoke-free policies” with “smoking norms” throughout the manuscript and updated the revised manuscript. Thanks for understanding in advance. 

R3-2: Authors have been including several variables, but do not provide any justification to include them. It seems more they have been “fishing” factors, without any clear theoretical background to consider specific variables. In the same line, the Discussion fails to provide explanations for why women, richer and more educated people live and work in more smoke-free environments.

Authors: Thanks again for asking this question and giving us an opportunity to improve the revised manuscript. We apologize for the unclear specifications of factors. Based on the previous literature, we have selected the input variables or factors. For the justification, we have added some references in the descriptions of the dataset section, and the findings of our significant factors are clearly explained in the discussion section. Thanks in advance for understanding. 

 R3-3: The correlation between SHS and smoke-free policies is not convincing as it does not consider potential confounders, as this is the case in the main analysis.

Authors: Thanks very much for your valuable comments. In this study, we try to only show the correlation between SHS and smoke-free policies. We do not try to consider SHS as a potential confounder of the smoke-free policies. As per your concerns, we will try to verify whether it is a potential confounder or not in our next projects. Thanks in advance for understanding. 

R3-4: There is a clear problem of reverse causation when using the smoker variable, as living in a more smoke-free environment reduces the likelihood to smoke. To avoid this problem, authors may use the prevalence of smoking in the region or municipality, e.g.

Authors: Thanks for asking this question. We are totally agreed with you. But, this study aimed to determine the impact of correlates on smoking norms both at home and workplaces in India. Without adjusting smoker variable in our main analysis, we could not find out the other correlates exactly. When we perform regression analysis, the independent variables asymptotically fixed and only dependent variable is considered as random variable. So, we think that it does not create any big problems for interpretation. Thanks in advance for understanding. 

R3-5: More generally, the implementation of norms is related to contextual variables, namely, the degree of enforcement of policies, cultural perceptions about smoking, other anti-tobacco policies (taxes, sales bans to minors, etc), or communication strategies by local authorities.

Authors: Thanks for your comments. We are totally agreed with you. We are already discussed in the discussion in the revised manuscript. We do not consider these variables in the main analysis because these variables are not available in our data. Thanks in advance for understanding. 

R3-6: The relevance of the paper is not clearly highlighted. Authors should state why measuring the determinants of smoke-free environments is relevant from a public health viewpoint, and how results may be relevant for decision making. We miss a section with the policy implications of the paper.

Authors: Thanks again for asking this question and giving us an opportunity to improve the revised manuscript. We already updated it’s in the revised manuscript. Thanks in advance for understanding. 

Dear Reviewer#3

Thank you very much for your encouragement, kind guidance, and support in helping the authors improve the manuscript. 

We are truly grateful for your time and support.

Kind Regards,

The Authors of the manuscript.

Reviewer (R4):

The authors have responded to all comments by the previous reviewers. However, some major language revision is still needed (especially the results and discussion sections) to help improve the readability of the manuscript.

Abstract

R4-1: Some of the sentences needs restructuring to make it read better.

Authors: Thank you very much for your valuable suggestions and giving us an opportunity to improve the revised manuscript. As per your suggestions, we have restructured some sentences in the abstract for the readers better understanding and updated the revised manuscript. Thanks, in advanced for understanding. 

 Introduction

R4-2: As mentioned by previous reviewers, please elaborate on the importance of the second objective (i.e. correlation between state-level smoke free policy and SHS exposure).

Authors: Thank you very much for your valuable suggestions and giving us an opportunity to improve the revised manuscript. As per your suggestions, we have updated the introduction section in the revised manuscript Thanks, in advanced for understanding. 

Results

R4-3: In many places, the sentences need to be rewritten. For example: Page 6 Line 15 “About 28.5% of tobacco smokers were kept to complete smoke-free policies in the home, whereas 48.5% of smokers in the workplace”. The meaning of this sentence is unclear. Do the authors mean “25% and 48.5% of tobacco smokers have smoke-free policies in their homes and workplace respectively?”

Authors: Thank you very much for your valuable suggestions and giving us an opportunity to improve the revised manuscript. As per your suggestions, we have restructured this sentence for readers clear understanding in result section and updated the revised manuscript. in the revised manuscript. Thanks again for understanding. 

Table 2:

R4-4: To indicate which variables were adjusted for in the multivariable model?

Authors: Thank you very much for your concerns and giving us an opportunity to improve the revised manuscript. In this study, we have applied multivariable or multiple logistic regression model. Suppose we have three independent factors, say A, B, and C which are significantly associated with dependent variable “Y”. These three factors are also considered as a significant factor under simple logistic regression (have only one independent variable with a dependent variable). In multivariable logistic regression model needs to done included all significant factors. Now, the odds ratio of independent factors “A” under simple logistic regression is unadjusted odds ratio while under multivariable logistic regression, it is adjusted odds ratio adjusting for the risk factor such as “B” and “C”.

In case of our study, we have 9 significant factors for smoking-free policies in home, get from Chi-Square test (p<0.05). Using these 9 significant factors, we fitted our multivariable logistic regression model and calculated the odds ratio. The odds ratio of “Tobacco smoking status” is called adjusted adjusting for the rests of 8 risk factors. We hope that it is clear now. Thanks again for understanding. 

R4-5: I suggest that the table be titled differently to better describe its contents. A suggestion is “Correlates of smoke-free policies in the home and the workplace in India”

Authors: Thank you very much for your valuable suggestions and giving us an opportunity to improve the revised manuscript. As per your suggestions, we have updated the entitled of the Table 2 as “Correlates of smoke-free policies in the home and the workplace in India”. Thanks again for understanding. 

R4-6: Th occupation ‘daily wise/causal labourer’ is probably meant to be ‘daily wage/casual labourer’?

Authors: Thank you very much for your valuable suggestions and giving us an opportunity to improve the revised manuscript. As per your suggestions, we have replaced the categories of occupations from ‘daily wise/causal labourer’ to ‘daily wage/casual labourer’ throughout the revised manuscript. Thanks again. 

Table 3:

R4-7: As pointed out by a previous reviewer, the word multivariable is appropriate here rather than multivariate logistic regression (The terms are not interchangeable).

Authors: Thank you very much for your valuable suggestions and giving us an opportunity to improve the revised manuscript. As per your suggestions, we have replaced the word from “multivariate logistic regression” to “multivariable logistic regression” throughout the revised manuscript. Thanks again. 

R4-8: ORs and the 95% CIs need be expressed up to 2 decimals only.

Authors: Thank again for your valuable suggestions and giving us an opportunity to improve the revised manuscript. As per your suggestions, we have added the value of ORs along with their 95% CI in 2 decimals throughout the revised manuscript. Thanks again. 

R4-9: The use of the ‘’registered’ ® symbol to indicate the reference category is rather inappropriate (Or does this symbol appear only on my computer?). The authors may put ‘1’, ‘Reference’ or ‘.ref’ in the AOR for the reference category instead to indicate this.

Authors: Thank again for your valuable suggestions and giving us an opportunity to improve the revised manuscript. As per your suggestions, we have fixed it and updated the revised manuscript. Thanks again. 

Table 4:

R4-10: The table should be given a more descriptive title e.g. “Correlations between smoke-free policies in the home and workplace and secondhanded smoke exposure”

Authors: Thank again for your valuable suggestions and giving us an opportunity to improve the revised manuscript. As per your suggestions, we have updated the entitled of the Table 4 as “Correlations between smoke-free policies in the home and workplace and secondhand smoke exposure”. Thanks again for understanding. 

R4-11: Only half of the table cells need to be filled as the other half is redundant (have the same values).

Authors: Thank again for your valuable suggestions and giving us an opportunity to improve the revised manuscript. As per your suggestions, we have fixed the Table 4 and updated the revised manuscript. Thanks again. 

R4-12: The correlation coefficient may be reported to only 2 decimals.

Authors: Thank again for your valuable suggestions and giving us an opportunity to improve the revised manuscript. As per your suggestions, we have kept the value of reported correlation coefficients in 2 decimals throughout the revised manuscript. Thanks again.

R4-13: ‘SHS at home’ should be ‘SHS exposure at home’ (similarly for workplace and public places).

Authors: Thank again for your valuable concerns. As per your suggestions, we have fixed these words throughout the manuscript these words and updated the revised manuscript. Thanks again for understanding. 

R4-14: In the footnote, SHS should stand for secondhand smoke instead of secondhand smoker.

Authors: Thank again for your valuable suggestions and giving us an opportunity to improve the revised manuscript. As per your suggestions, we have modified the footnote of Table 4 and updated the revised manuscript. Thanks again for understanding. 

Discussion

Suggest to first summarize the finding and then interpretation and discussion in each paragraph. For example on Pg 8 para 3, it would help to state the correlation finding first before explaining the possible reasons why it is so.

The language style throughout the section need to be given a another round of proofreading.

- Example on Pg 8 “Literature suggests the fact that” should be “literature suggest that”

Pg 8, “Other ocupations tend to..” does “other occupations” mean non-professional occupations?

Pg 8: The statement “unspoiled , contaminated environment”, unspoiled and contaminated are contradictory, please check this statement.

Authors: Thank again for your valuable suggestions and giving us an opportunity to improve the revised manuscript. As per your suggestions, we have rewritten the discussion section and update the revised manuscript. Thanks in advance. 

Conclusion

The implications of the study findings with regards to tobacco control policy as commented by the previous reviewer are still not spelled out. What can the authorities do with the findings? Make more specific suggestions/recommendations.

Authors: Thank again for your valuable suggestions and giving us an opportunity to improve the revised manuscript. As per your suggestions, we have rewritten the conclusion section and update the revised manuscript. Thanks in advance. 

Dear Reviewer#4

Thank you very much for your encouragement, kind guidance, and support in helping the authors improve the manuscript. 

We are truly grateful for your time and support.

Kind Regards,

The Authors of the manuscript.

---

## [Decision Letter · Decision Letter 3]

5 Jul 2022

PONE-D-20-41034R3Smoking norms in the home and in the workplace among Indian people: Evidence from Global Adult Tobacco Survey Data-2016/2017PLOS ONE

Dear Dr. Alauddin,

Thank you for submitting your manuscript to PLOS ONE. After careful consideration, we feel that it has merit but does not fully meet PLOS ONE’s publication criteria as it currently stands. Therefore, we invite you to submit a revised version of the manuscript that addresses the points raised during the review process.

The manuscript has been evaluated by two reviewers, and their comments are available below.

The reviewers have raised a number of major concerns. They feel the manuscript should outline a clearly-defined research question, and they request improvements to the reporting of methodological aspects of the study. They also request improvement to the English language quality.

Could you please carefully revise the manuscript to address all comments raised?

We look forward to receiving your revised manuscript.

Kind regards,

Thomas Phillips, PhD

Staff Editor

PLOS ONE

Reviewers' comments:

Reviewer's Responses to Questions

**Comments to the Author**

1. If the authors have adequately addressed your comments raised in a previous round of review and you feel that this manuscript is now acceptable for publication, you may indicate that here to bypass the “Comments to the Author” section, enter your conflict of interest statement in the “Confidential to Editor” section, and submit your "Accept" recommendation.

Reviewer #3: (No Response)

Reviewer #4: All comments have been addressed

2. Is the manuscript technically sound, and do the data support the conclusions?

Reviewer #3: No

Reviewer #4: Yes

3. Has the statistical analysis been performed appropriately and rigorously? 

Reviewer #3: Yes

Reviewer #4: Yes

4. Have the authors made all data underlying the findings in their manuscript fully available?

Reviewer #3: Yes

Reviewer #4: Yes

5. Is the manuscript presented in an intelligible fashion and written in standard English?

Reviewer #3: No

Reviewer #4: No

6. Review Comments to the Author

Reviewer #3: (No Response)

Reviewer #4: 1) I beg to differ with the other reviewer who suggested changing the term "smoke-free policy" to "smoking norms". In my opinion, using the term "smoke-free policy" is more accurate.

2) In Table 3, please place the number 1 to indicate the reference category in the AOR column instead of the p value column.

3)All other issues were addressed except for language, which still needs correcting.

Example: In Discussion, Para 3- "Literatures are suggesting the fact........ than people engaged in other government/non government professional careers" needs to be rewritten. In its current form, the meaning of this sentence is unclear.

7. PLOS authors have the option to publish the peer review history of their article (what does this mean?). If published, this will include your full peer review and any attached files.

Reviewer #3: No

Reviewer #4: No

---

## [Author Response · Author response to Decision Letter 3]

20 Jul 2022

Authors’ Response to Reviewers

Manuscript ID: PONE-D-20-41034R3

Title: Smoke-free policies in the home and in the workplace among Indian people: Evidence from Global Adult Tobacco Survey Data-2017

Dear Respected Editor-in-Chief,

First of all, we thank the reviewer and the editor for their useful suggestions for improving the manuscript. We have now revised the manuscript accordingly. Thank you once again and looking forward to the final acceptance in your esteemed journal.

Reviewer (R3):

The paper has been improved but important limitations remain:

(R3-1): We still miss a justification of the variables’ choice. Mentioning that they have been used in the literature is not enough. We must understand which mechanisms may lead subgroups to adopt stricter norms or not. In the Discussion, no good explanations are presented for some groups adopting stricter norms (e.g., more educated or richer people).

Authors: Thank you for your valuable suggestions and giving us an opportunity to improve the revised manuscript. We try to add the justification of including various explanatory variables for this research work and modify the discussion section to include the possible reasons for adopting stricter norms among some specified groups (e.g., more educated or richer people).

(R3-2): Since this information is missing about potential causal mechanisms, the Discussion does not provide relevant insights for decision making.

Authors: Thanks again for your valuable suggestions. We have focused on this suggestion and rewritten some points in the discussion section to make the potential causal mechanisms clear. Thanks for understanding in advance. 

(R3-2): The text needs a careful revision of English language.

Authors: Thank you for your comment. We have given a special attention to maintaining the standard English language for this article. Thanks for understanding in advance. 

Dear Reviewer#3

Thank you very much for your encouragement, kind guidance, and support in helping the authors improve the manuscript. 

We are truly grateful for your time and support.

Kind Regards,

The Authors of the manuscript.

Reviewer (R4):

(R4-1): I beg to differ with the other reviewer who suggested changing the term "smoke-free policy" to "smoking norms". In my opinion, using the term "smoke-free policy" is more accurate.

Authors: We highly appreciate your valuable comments. We have changed the term "smoke-free policy" to "smoking norms" throughout the manuscript. Thanks for understanding in advance. 

(R4-2): In Table 3, please place the number 1 to indicate the reference category in the AOR column instead of the p-value column.

Authors: Thank you for your valuable suggestions. We regret making such a great mistake. Now, we have fixed it and updated the revised manuscript. Thanks again.

(R4-3): All other issues were addressed except for language, which still needs correcting. Example: In Discussion, Para 3-"Literatures are suggesting the fact than people engaged in other government/non government professional careers" needs to be rewritten. In its current form, the meaning of this sentence is unclear.

Authors: Thank you for your opinion. This point has been carefully addressed and updated in the revised manuscript. Moreover, we have given special attention to maintaining the standard English language for this article. Thanks again for your undderstadning in advance. 

Dear Reviewer#4

Thank you very much for your encouragement, kind guidance, and support in helping the authors improve the manuscript. 

We are truly grateful for your time and support.

Kind Regards,

The Authors of the manuscript.

---

## [Decision Letter · Decision Letter 4]

17 Oct 2022

PONE-D-20-41034R4Smoke-free policies in the home and in the workplace among Indian people: Evidence from Global Adult Tobacco Survey Data-2016/2017PLOS ONE

Dear Dr. Alauddin,

Thank you for submitting your manuscript to PLOS ONE. After careful consideration, we feel that it has merit but does not fully meet PLOS ONE’s publication criteria as it currently stands. Therefore, we invite you to submit a revised version of the manuscript that addresses the points raised during the review process.

We look forward to receiving your revised manuscript.

Kind regards,

Sandra Boatemaa Kushitor, Ph.D.

Academic Editor

PLOS ONE

Additional Editor Comments:

Dear Authors,

This manuscript has potential but cannot be accepted for publication in its current format. The authors change the focus of the manuscript at different points in the manuscript. The written is unclear. The authors should confirm whether they are examining the influence of smoke-free policies on smoke-free practices at home and work, and whether adopting smoke-free practices result in less SHS. I believe this is the story they want to tell. However, it is unclear. They should also specify whether it is the same people they they assessed for smoke-free practices at home and at work. Sometimes the difference is unclear in their write up.

Reviewers' comments:

Reviewer's Responses to Questions

**Comments to the Author**

1. If the authors have adequately addressed your comments raised in a previous round of review and you feel that this manuscript is now acceptable for publication, you may indicate that here to bypass the “Comments to the Author” section, enter your conflict of interest statement in the “Confidential to Editor” section, and submit your "Accept" recommendation.

Reviewer #5: (No Response)

Reviewer #6: All comments have been addressed

2. Is the manuscript technically sound, and do the data support the conclusions?

Reviewer #5: Partly

Reviewer #6: Yes

3. Has the statistical analysis been performed appropriately and rigorously? 

Reviewer #5: Yes

Reviewer #6: Yes

4. Have the authors made all data underlying the findings in their manuscript fully available?

Reviewer #5: Yes

Reviewer #6: Yes

5. Is the manuscript presented in an intelligible fashion and written in standard English?

Reviewer #5: No

Reviewer #6: Yes

6. Review Comments to the Author

Reviewer #5: Overall very useful topic, but these substantial revisions, and language editing.

Introduction

The prevalence of male and female tobacco consumers is 42.4% and 14.2%. The denominator is not clear here.

References 6 and 13 in page number 3. How do these references say that there is progress in smoke-free policies in indoor public/workplaces

The lack of evidence on smoke-free policies in homes and suboptimal implementation of smoke-free workplaces need to be highlighted to provide a rationale for doing this study.

Page number 05: “It is classified into binary categories in the following ways: the respondent was classified as living in a completely smoke-free home”

Living in a smoke-free home is a misnomer here as we don’t know whether the home is smoke-free or not. We only know that there is smoke-free policy in the home. So you have to choose these words carefully throughout the manuscript.

So we can say that the categories are: having a smoke-free policy at home or not?

Table 3: Please put p-value column on the right side of the aOR column

Table 4: please clearly state that these correlations depict relationships at a state level and also write the total number (N) for each correlation.

Major comments:

There is a major confusion in this paper. You are defining two things in the methods section (variables of the study sub-section): smoke-free policy in home and workplace and exposure to SHS in home and workplace. Both are different concepts as rightly defined.

However, in the results you are probably exploring correlates of smoke-free policy. Why do you define SHS exposure under the variables of the study section?

Out of 74, 037 we chose 71,046 and 15,254 for SHS at home and workplace respectively. How did you arrive at these numbers? Please provide a flowchart showing the flow of respondents.

Discussion

65 and above are mostly tended to stay smoke-free in their home h. Please rectify this sentence

“female people emphasized more on a smoke-free zone in their workplace more than their opposite gender”

“Married people were less tending to enjoy a complete non-smoking work environment as well as their home than their single counterparts”

“Contrarily, separated persons were expected to enjoy a more smokefree environment in their residence.”

The use of verbs such as emphasize, tending to enjoy, expected to enjoy does not seem to fit your findings. Rephrasing these statements would help.

Muslims were more likely to maintain a smoke-free workplace but not a smoke-free home. The use of religion as an explanation seems contradicting here.

Any possible reason/speculation for the regional differences in smoke-free policies? Exploring the answers and possible speculations require a thorough understanding of the country’s culture, tobacco habits, regional differences which might not be the case here as all the authors belong to another country i.e. Bangladesh. I wonder what is the rationale for conducting this study in an Indian setting.

Any possible reason/speculation for the association of education and occupation with Smoke free policy.

Discussion section lacks a paragraph on policy implications based on the key findings of the study

“However, we didn’t adjust some other possible confounding factors which could be explored further”

Mention what are the possible confounders which could be explored further

Page 09: Furthermore, if an individual is forced to be refrained from smoking in the workplace, it would ultimately lead to quit smoking in the home also. Give suitable references here

Discussion needs more streamlining. It should begin with the key highlights of the paper and each highlight could be discussed in separate paragraph with reference to other studies in the literature, possible reasons and speculations or differences within studies.

Conclusion looks a bit repetitive of what has been said before. A more crisp and clear messaging would help.

Reviewer #6: This study examines factors associated with smoke-free policies at home and in workplace using the 2017 Global Adult Tobacco Survey (GATS) in India. Additionally, the study examines the correlations between secondhand smoke exposure and smokefree policies. The variables were selected from the literature; therefore, the findings were mostly consistent with the existing literature. Nevertheless, this is the first kind of study in a country with the second largest of tobacco users in the world. The main results show that substantial number of people in the country are not covered by complete smokefree policies which is significant to inform policy initiatives. As such, it provides an added value to the literature. Further, the authors appear to be responsive to earlier reviews.

With the above said, here are few suggestions:

1) Consider creating a third paragraph for the introduction, starting with "Several studies have been conducted ...."

2) The study's aim is still not well-articulated in the paragraph #2.

3) In section 2.1, did you examine that the data you excluded were not significantly different from the analytic samples.

4) In section 2.4, add the name of the manufacturer and the City and country in a parenthesis after "STATA version 12 (...., ....)

5) In reporting the results (section 3.2), can you report OR, CI, and P-value?

6) Please note that this is a cross-sectional study; therefore, the results should not be interpreted as establishing a causation, instead of association

7) Key limitations of the study are missing, particularly biases such as recall and social desirability. The cross-sectional data by itself is a limitation

The minor issues involve attention to details ---

1) Place comma (,) before "respectively"

2) Delete "h" on the third line under section 4 Discussion

3) Consider removing grid on tables 2 and 3

7. PLOS authors have the option to publish the peer review history of their article (what does this mean?). If published, this will include your full peer review and any attached files.

Reviewer #5: **Yes: **Jaya Prasad Tripathy

Reviewer #6: No

---

## [Author Response · Author response to Decision Letter 4]

9 Dec 2022

Reviewer (R5)

Overall very useful topic, but these substantial revisions, and language editing.

R5-1: The prevalence of male and female tobacco consumers is 42.4% and 14.2%. The denominator is not clear here.

Authors: Thank you for your comment. As per your concerns, we modified this sentence for more clear understanding and cited its supporting documents. Thanks again for understanding. 

R5-2: References 6 and 13 in page number 3. How do these references say that there is progress in smoke-free policies in indoor public/workplaces

Authors: Thank you very much for raising this question. In reference 6 there is section of “Change in exposure to secondhand smoke at various places; GATS 1 to GATS 2” where we noticed the change in exposure to secondhand smoke at different public places among all adults, from GATS 1 India, 2009-10 to GATS 2 India, 2016-17. This difference in SHS exposure from GATS 1 to GATS 2 data may reflect the impact of law enforcement and increased public awareness in support of smoke-free policy. It indicates the progress of smoke free policy across the country. Based on that evidence this reference is placed here. We hope this will support the justification of using this reference. 

R5-3: The lack of evidence on smoke-free policies in homes and suboptimal implementation of smoke-free workplaces need to be highlighted to provide a rationale for doing this study.

Authors: Thank you very much for your kind suggestion. We have included some information to provide justification and evidence on suboptimal implementation of smoke-free policy both at home and workplaces to highlight the rationale of the study. 

R5-4: Page number 05: “It is classified into binary categories in the following ways: the respondent was classified as living in a completely smoke-free home”

Living in a smoke-free home is a misnomer here as we don’t know whether the home is smoke-free or not. We only know that there is smoke-free policy in the home. So, you have to choose these words carefully throughout the manuscript. So, we can say that the categories are: having a smoke-free policy at home or not?

Authors: Thank you so much for your valuable suggestion. As per your suggestions, we have modified this statement in the revised manuscript. Thanks again for understanding. 

R5-5: Table 3: Please put p-value column on the right side of the a OR column. 

Authors: Thank again for your valuable suggestion. As per your suggestion, we put the p-vale column of the right side of OR column and modified the Table 3 in the revised manuscript. Thanks again for understanding. 

R5-6: Table 4: please clearly state that these correlations depict relationships at a state level and also write the total number (N) for each correlation.

Authors: Thank you very much for your valuable comments and concerns. As per your concerns, we put the total number (N) for each correction in Table and also modified Table 4 in the revised manuscript. Thanks again for understanding. 

R5-7: Major comments: There is a major confusion in this paper. You are defining two things in the methods section (variables of the study sub-section): smoke-free policy in home and workplace and exposure to SHS in home and workplace. Both are different concepts as rightly defined. However, in the results you are probably exploring correlates of smoke-free policy. Why do you define SHS exposure under the variables of the study section?

Authors: Thank you very much for giving your valuation concerns. As per your concerns, we move this part from here and make another subsection as study or dependent variable under methodology section in the revised manuscript. Yes, we estimated the state-level correlation between the proportion of smoke-free policies and SHS exposure at both home and workplace. To make people understand what SHS is and how it is defined, it is added to the study section.

 R5-8: Out of 74, 037 we chose 71,046 and 15,254 for SHS at home and workplace respectively. How did you arrive at these numbers? Please provide a flowchart showing the flow of respondents.

Authors: Thank you very much for your valuable concerns. As per your concerns, we added a flowchart for both SHS at home and workplace in the revised manuscript. Thanks again for understanding.

R5-9: Discussion: 65 and above are mostly tended to stay smoke-free in their home h. Please rectify this sentence. 

Authors: Thank you very much for your valuable comments. It was typing mistake. We apologized for such a mistake. This sentence is corrected and updated in the revised manuscript. Thanks again for understanding. 

R510: “female people emphasized more on a smoke-free zone in their workplace more than their opposite gender”

“Married people were less tending to enjoy a complete non-smoking work environment as well as their home than their single counterparts”

“Contrarily, separated persons were expected to enjoy a more smokefree environment in their residence.”

The use of verbs such as emphasize, tending to enjoy, expected to enjoy does not seem to fit your findings. Rephrasing these statements would help.

Authors: Thank you very much for your valuable suggestions. As per your suggestions, all these terms have been removed from the discussion section in order to make the statements more perceptible. Thanks again for understanding. 

R5-11: Muslims were more likely to maintain a smoke-free workplace but not a smoke-free home. The use of religion as an explanation seems contradicting here.

Any possible reason/speculation for the regional differences in smoke-free policies? Exploring the answers and possible speculations require a thorough understanding of the country’s culture, tobacco habits, regional differences which might not be the case here as all the authors belong to another country i.e. Bangladesh. I wonder what is the rationale for conducting this study in an Indian setting.

Authors: Thank you very much for comments. Yes, you are right. In our previous version revised manuscript, we mistakenly written this statement. We are extremely sorry regarding this issue. Actually, we got only other religion people as a significant risk factors compared to Hindu religions. for completely smoke-free policy at home. In this work, we observed that other religious people were 1.939 [AOR=1.939, 95% CI= 1.65 to 2.28, p<0.001] times more likely to follow complete smoke-free policy at home compared to Hindu religious people. As per your concerns, we fixed this statement in the discussion section of the revised manuscript. Thanks again for understanding. 

R5-12: Any possible reason/speculation for the association of education and occupation with Smoke free policy. Discussion section lacks a paragraph on policy implications based on the key findings of the study

Authors: Thank you very much for noticing this point. We have tried to highlight the possible reasons of association for education and occupation with Smoke free policy. Besides, based on the study findings, possible policy implications are added to the conclusion section. Thanks again for understanding. 

R5-13: “However, we didn’t adjust some other possible confounding factors which could be explored further”. Mention what are the possible confounders which could be explored further.

Authors: Thank you very much for your comment and valuable concerns. As per your concerns, we already mentioned some possible confounding factors which could be explored in future. Thanks again. 

R5-14: Page 09: Furthermore, if an individual is forced to be refrained from smoking in the workplace, it would ultimately lead to quit smoking in the home also. Give suitable references here

Authors: Thank you very much for your valuable concerns. As per your concerns, we provided a suitable citation in order to support this statement. Thanks again for understanding. 

 R5-15: Discussion needs more streamlining. It should begin with the key highlights of the paper and each highlight could be discussed in separate paragraph with reference to other studies in the literature, possible reasons and speculations or differences within studies.

Authors: Thank you very much for your valuable suggestion. As per your suggestions, we have updated the discussion section to make the potential mechanism clear in the revised manuscript. Thanks again for understanding. 

R5-16: Conclusion looks a bit repetitive of what has been said before. A more crisp and clear messaging would help.

Authors: Thanks again for your valuable suggestion. As per your suggestions, we updated this section and focused on some possible policy implications here. 

Dear Reviewer#5

Thank you very much for your encouragement, kind guidance, and support in helping the authors improve the manuscript.

We are truly grateful for your time and support. Kind Regards,

The Authors of the manuscript.

Reviewer (R6)

This study examines factors associated with smoke-free policies at home and in workplace using the 2017 Global Adult Tobacco Survey (GATS) in India. Additionally, the study examines the correlations between secondhand smoke exposure and smokefree policies. The variables were selected from the literature; therefore, the findings were mostly consistent with the existing literature. Nevertheless, this is the first kind of study in a country with the second largest of tobacco users in the world. The main results show that substantial number of people in the country are not covered by complete smokefree policies which is significant to inform policy initiatives. As such, it provides an added value to the literature. Further, the authors appear to be responsive to earlier reviews. With the above said, here are few suggestions:

R6-1: Consider creating a third paragraph for the introduction, starting with "Several studies have been conducted ...."

Authors: Thank you for your suggestion. As per your suggestion, we have created a new paragraph from the given statement. Thanks again for understanding. 

 R6-2: The study's aim is still not well-articulated in the paragraph #2.

Authors: Thank you very much for asking this question and giving us an opportunity to improve the revised manuscript. As per your concerns, we are now explained more clearly the objective of this study. Thanks again for understanding. 

R6-3: In section 2.1, did you examine that the data you excluded were not significantly different from the analytic samples.

Authors: Thank you very much for raising this question. In this work, we have excluded some observations, like don’t know, refused, and missing values during the analysis. We think that these observations are not statistically significant. As you’re your concerns, we have added a flowchart for the selection of samples in the revised manuscript (See in Figure 1). Thanks again for understanding. 

R6-4: In section 2.4, add the name of the manufacturer and the City and country in a parenthesis after "STATA version 12 (...., ....)

Authors: Thank you very much for your valuable comments. As per concerns, we added STATA version along with developer names as well as name of the manufacturer and the City and country in order to reduce the results. Thanks again for understanding. 

R6-5: In reporting the results (section 3.2), can you report OR, CI, and P-value?

Authors: Thank you very much for your valuable suggestion. As per your suggestion, we have added the value of OR with its CI and p-value and explained more clearly in the section 3.2 of the revised manuscript. Thanks again for understanding. 

R6-6: Please note that this is a cross-sectional study; therefore, the results should not be interpreted as establishing a causation, instead of association.

Authors: Thank you very much for your comment. Yes, this is a cross-sectional study. As per your suggestions, we have fixed it in the revised manuscript. Thanks again for understanding. 

R6-7: Key limitations of the study are missing, particularly biases such as recall and social desirability. The cross-sectional data by itself is a limitation

Authors: Thank you very much for your comment. As per your comments, we have added section as “Strength and limitations” in the revised manuscript. Thanks again for understanding. 

R6-8: The minor issues involve attention to details ---

1) Place comma (,) before "respectively"

2) Delete "h" on the third line under section 4 Discussion

3) Consider removing grid on tables 2 and 3

Authors: We highly appreciate your valuable comments. As per your concerns, all points are addressed carefully and updated the revised manuscript. Thanks again for understanding. 

Dear Reviewer#6

Thank you very much for your encouragement, kind guidance, and support in helping the authors improve the manuscript.

We are truly grateful for your time and support. Kind Regards,

The Authors of the manuscript.

---

## [Editor Report · Decision Letter 5]

13 Dec 2022

PONE-D-20-41034R5Smoke-free policies in the home and in the workplace among Indian people: Evidence from Global Adult Tobacco Survey Data-2016/2017PLOS ONE

Dear Dr. Alauddin,

Thank you for submitting your manuscript to PLOS ONE. After careful consideration, we feel that it has merit but does not fully meet PLOS ONE’s publication criteria as it currently stands. Therefore, we invite you to submit a revised version of the manuscript that addresses the points raised during the review process.

Before your manuscript can be consider for review, please respond to the comments of the editor. The comments can be found in the attached pdf document.  Please submit your revised manuscript by Jan 27 2023 11:59PM. If you will need more time than this to complete your revisions, please reply to this message or contact the journal office at plosone@plos.org. Please include the following items when submitting your revised manuscript:A rebuttal letter that responds to each point raised by the academic editor and reviewer(s). You should upload this letter as a separate file labeled 'Response to Reviewers'.A marked-up copy of your manuscript that highlights changes made to the original version. You should upload this as a separate file labeled 'Revised Manuscript with Track Changes'.An unmarked version of your revised paper without tracked changes. You should upload this as a separate file labeled 'Manuscript'.

We look forward to receiving your revised manuscript.

Kind regards,

Sandra Boatemaa Kushitor, Ph.D.

Academic Editor

PLOS ONE

Additional Editor Comments:

Dear Authors,

Please respond to the comments raised by the editor and provide a rebuttal these comments in the response to reviewers file. My comments are in the reattached pdf.

Thank you.

---

## [Author Response · Author response to Decision Letter 5]

14 Dec 2022

Authors’ Response to Reviewers

Manuscript ID: PONE-D-20-41034R3+R4

Title: Smoke-free status of homes and workplaces among Indian people: Evidence from Global Adult Tobacco Survey Data-2016/2017

Dear Respected Editor-in-Chief,

We would like to express our gratitude and thanks to the reviewer and the editor for their useful suggestions for improving the manuscript. We have revised the manuscript accordingly. Thank you once again and looking forward to the final acceptance in your esteemed journal.

Editor-in-Chief (EC)

EC1: In title, “Smoke-free policies in the home and in the workplace among Indian people” should be “Smoke-free status of homes and workplaces among Indian people: Evidence…..”

Authors: Thank you very much for your comment. As per your concerns, we have modified the title for clear understanding. Thanks again for understanding. 

E2: In introduction, 1st paragraph, Tobacco? You can take smoke edit for clarity.

Authors: Thank you very much for addressing this point and your valuable suggestions. As per your suggestions, we removed the term ‘take smoke’ and modified this sentence. Thanks again for understanding. 

E3: In Section 2.1, Use “GATS 2 conducted” instead of “This study conducted”

Authors: Thank you for your kind suggestion. As per your suggestions, we changed the term ‘this study’ to ‘GATS2’. Thank you once again for understanding. 

E4: The sample size for workplace and home varies. are you able to assess whether a person has is smoke free both at home and at the workplace? If not, your results seem to imply this. Please check and correct

Authors: Thank you very much for raising this question and your valuable concerns. As per your concerns, we added a flowchart for sample size selection for both SHS at home and workplace in the revised manuscript (See in Figure 1). Thanks again for understanding.

E5: Use Practices instead of policies

Authors: Thank you for your valuable suggestions. As per your suggestions, we have changed the term ‘policies’ to ‘practice’. Thank you for understanding.

E6: Use “Principal component analysis (PCA) was conducted” instead of “Principal component analysis (PCA) has been conducted”

 Authors: Thank you for your valuable concerns. As per your suggestions, we have fixed this sentence and updated the revised manuscript. Thank you for understanding.

E7: In Section 3.2, Replace with factors associated with smoke free practices at work and home with Logistic Regression Analysis.

Authors: Thank you for your kind concern. We have changed the title of this section considering your valuable suggestion. Thank you once again. 

E8: Is SHS exposure your outcome variable. That’s not how you have positioned this paper from the introduction and methods.

Authors: Thank you very much for your valuable concerns. We have tried to make clear that the association between secondhand smoker and smoke free policies is an additional part of the analysis to understand their linkage. We have added a line in the introduction section for a clear understanding. Also, in the methodology section, we have separated the definition of SHS from the dependent variable section (Section 2.3.1) to avoid the confusion. We hope that will be helpful. Thank you for understanding. 

E9: Why are these highlighted

Authors: Thank you for pointing this out. We apologize for such a mistake and already fixed its. Thanks again for understanding. 

E10: Space is missing

Authors: Thank you for noticing this. As per your concerns, it is addressed carefully. Thank you for understanding. 

Dear Editor-in-Chief

Thank you very much for your encouragement, kind guidance, and support in helping the authors improve the manuscript.

We are truly grateful for your time and support. Kind Regards,

The Authors of the manuscript.

Reviewer (R5)

Overall very useful topic, but these substantial revisions, and language editing.

R5-1: The prevalence of male and female tobacco consumers is 42.4% and 14.2%. The denominator is not clear here.

Authors: Thank you for your comment. As per your concerns, we modified this sentence for more clear understanding and cited its supporting documents. Thanks again for understanding. 

R5-2: References 6 and 13 in page number 3. How do these references say that there is progress in smoke-free policies in indoor public/workplaces

Authors: Thank you very much for raising this question. In reference 6 there is section of “Change in exposure to secondhand smoke at various places; GATS 1 to GATS 2” where we noticed the change in exposure to secondhand smoke at different public places among all adults, from GATS 1 India, 2009-10 to GATS 2 India, 2016-17. This difference in SHS exposure from GATS 1 to GATS 2 data may reflect the impact of law enforcement and increased public awareness in support of smoke-free policy. It indicates the progress of smoke free policy across the country. Based on that evidence this reference is placed here. We hope this will support the justification of using this reference. 

R5-3: The lack of evidence on smoke-free policies in homes and suboptimal implementation of smoke-free workplaces need to be highlighted to provide a rationale for doing this study.

Authors: Thank you very much for your kind suggestion. We have included some information to provide justification and evidence on suboptimal implementation of smoke-free policy both at home and workplaces to highlight the rationale of the study. 

R5-4: Page number 05: “It is classified into binary categories in the following ways: the respondent was classified as living in a completely smoke-free home”

Living in a smoke-free home is a misnomer here as we don’t know whether the home is smoke-free or not. We only know that there is smoke-free policy in the home. So, you have to choose these words carefully throughout the manuscript. So, we can say that the categories are: having a smoke-free policy at home or not?

Authors: Thank you so much for your valuable suggestion. As per your suggestions, we have modified this statement in the revised manuscript. Thanks again for understanding. 

R5-5: Table 3: Please put p-value column on the right side of the a OR column. 

Authors: Thank again for your valuable suggestion. As per your suggestion, we put the p-vale column of the right side of OR column and modified the Table 3 in the revised manuscript. Thanks again for understanding. 

R5-6: Table 4: please clearly state that these correlations depict relationships at a state level and also write the total number (N) for each correlation.

Authors: Thank you very much for your valuable comments and concerns. As per your concerns, we put the total number (N) for each correction in Table and also modified Table 4 in the revised manuscript. Thanks again for understanding. 

R5-7: Major comments: There is a major confusion in this paper. You are defining two things in the methods section (variables of the study sub-section): smoke-free policy in home and workplace and exposure to SHS in home and workplace. Both are different concepts as rightly defined. However, in the results you are probably exploring correlates of smoke-free policy. Why do you define SHS exposure under the variables of the study section?

Authors: Thank you very much for giving your valuation concerns. As per your concerns, we move this part from here and make another subsection as study or dependent variable under methodology section in the revised manuscript. Yes, we estimated the state-level correlation between the proportion of smoke-free policies and SHS exposure at both home and workplace. To make people understand what SHS is and how it is defined, it is added to the study section.

 R5-8: Out of 74, 037 we chose 71,046 and 15,254 for SHS at home and workplace respectively. How did you arrive at these numbers? Please provide a flowchart showing the flow of respondents.

Authors: Thank you very much for your valuable concerns. As per your concerns, we added a flowchart for both SHS at home and workplace in the revised manuscript. Thanks again for understanding.

R5-9: Discussion: 65 and above are mostly tended to stay smoke-free in their home h. Please rectify this sentence. 

Authors: Thank you very much for your valuable comments. It was typing mistake. We apologized for such a mistake. This sentence is corrected and updated in the revised manuscript. Thanks again for understanding. 

R510: “female people emphasized more on a smoke-free zone in their workplace more than their opposite gender”

“Married people were less tending to enjoy a complete non-smoking work environment as well as their home than their single counterparts”

“Contrarily, separated persons were expected to enjoy a more smokefree environment in their residence.”

The use of verbs such as emphasize, tending to enjoy, expected to enjoy does not seem to fit your findings. Rephrasing these statements would help.

Authors: Thank you very much for your valuable suggestions. As per your suggestions, all these terms have been removed from the discussion section in order to make the statements more perceptible. Thanks again for understanding. 

R5-11: Muslims were more likely to maintain a smoke-free workplace but not a smoke-free home. The use of religion as an explanation seems contradicting here.

Any possible reason/speculation for the regional differences in smoke-free policies? Exploring the answers and possible speculations require a thorough understanding of the country’s culture, tobacco habits, regional differences which might not be the case here as all the authors belong to another country i.e. Bangladesh. I wonder what is the rationale for conducting this study in an Indian setting.

Authors: Thank you very much for comments. Yes, you are right. In our previous version revised manuscript, we mistakenly written this statement. We are extremely sorry regarding this issue. Actually, we got only other religion people as a significant risk factors compared to Hindu religions. for completely smoke-free policy at home. In this work, we observed that other religious people were 1.939 [AOR=1.939, 95% CI= 1.65 to 2.28, p<0.001] times more likely to follow complete smoke-free policy at home compared to Hindu religious people. As per your concerns, we fixed this statement in the discussion section of the revised manuscript. Thanks again for understanding. 

R5-12: Any possible reason/speculation for the association of education and occupation with Smoke free policy. Discussion section lacks a paragraph on policy implications based on the key findings of the study

Authors: Thank you very much for noticing this point. We have tried to highlight the possible reasons of association for education and occupation with Smoke free policy. Besides, based on the study findings, possible policy implications are added to the conclusion section. Thanks again for understanding. 

R5-13: “However, we didn’t adjust some other possible confounding factors which could be explored further”. Mention what are the possible confounders which could be explored further.

Authors: Thank you very much for your comment and valuable concerns. As per your concerns, we already mentioned some possible confounding factors which could be explored in future. Thanks again. 

R5-14: Page 09: Furthermore, if an individual is forced to be refrained from smoking in the workplace, it would ultimately lead to quit smoking in the home also. Give suitable references here

Authors: Thank you very much for your valuable concerns. As per your concerns, we provided a suitable citation in order to support this statement. Thanks again for understanding. 

 R5-15: Discussion needs more streamlining. It should begin with the key highlights of the paper and each highlight could be discussed in separate paragraph with reference to other studies in the literature, possible reasons and speculations or differences within studies.

Authors: Thank you very much for your valuable suggestion. As per your suggestions, we have updated the discussion section to make the potential mechanism clear in the revised manuscript. Thanks again for understanding. 

R5-16: Conclusion looks a bit repetitive of what has been said before. A more crisp and clear messaging would help.

Authors: Thanks again for your valuable suggestion. As per your suggestions, we updated this section and focused on some possible policy implications here. 

Dear Reviewer#5

Thank you very much for your encouragement, kind guidance, and support in helping the authors improve the manuscript.

We are truly grateful for your time and support. Kind Regards,

The Authors of the manuscript.

Reviewer (R6)

This study examines factors associated with smoke-free policies at home and in workplace using the 2017 Global Adult Tobacco Survey (GATS) in India. Additionally, the study examines the correlations between secondhand smoke exposure and smokefree policies. The variables were selected from the literature; therefore, the findings were mostly consistent with the existing literature. Nevertheless, this is the first kind of study in a country with the second largest of tobacco users in the world. The main results show that substantial number of people in the country are not covered by complete smokefree policies which is significant to inform policy initiatives. As such, it provides an added value to the literature. Further, the authors appear to be responsive to earlier reviews. With the above said, here are few suggestions:

R6-1: Consider creating a third paragraph for the introduction, starting with "Several studies have been conducted ...."

Authors: Thank you for your suggestion. As per your suggestion, we have created a new paragraph from the given statement. Thanks again for understanding. 

 R6-2: The study's aim is still not well-articulated in the paragraph #2.

Authors: Thank you very much for asking this question and giving us an opportunity to improve the revised manuscript. As per your concerns, we are now explained more clearly the objective of this study. Thanks again for understanding. 

R6-3: In section 2.1, did you examine that the data you excluded were not significantly different from the analytic samples.

Authors: Thank you very much for raising this question. In this work, we have excluded some observations, like don’t know, refused, and missing values during the analysis. We think that these observations are not statistically significant. As you’re your concerns, we have added a flowchart for the selection of samples in the revised manuscript (See in Figure 1). Thanks again for understanding. 

R6-4: In section 2.4, add the name of the manufacturer and the City and country in a parenthesis after "STATA version 12 (...., ....)

Authors: Thank you very much for your valuable comments. As per concerns, we added STATA version along with developer names as well as name of the manufacturer and the City and country in order to reduce the results. Thanks again for understanding. 

R6-5: In reporting the results (section 3.2), can you report OR, CI, and P-value?

Authors: Thank you very much for your valuable suggestion. As per your suggestion, we have added the value of OR with its CI and p-value and explained more clearly in the section 3.2 of the revised manuscript. Thanks again for understanding. 

R6-6: Please note that this is a cross-sectional study; therefore, the results should not be interpreted as establishing a causation, instead of association.

Authors: Thank you very much for your comment. Yes, this is a cross-sectional study. As per your suggestions, we have fixed it in the revised manuscript. Thanks again for understanding. 

R6-7: Key limitations of the study are missing, particularly biases such as recall and social desirability. The cross-sectional data by itself is a limitation

Authors: Thank you very much for your comment. As per your comments, we have added section as “Strength and limitations” in the revised manuscript. Thanks again for understanding. 

R6-8: The minor issues involve attention to details ---

1) Place comma (,) before "respectively"

2) Delete "h" on the third line under section 4 Discussion

3) Consider removing grid on tables 2 and 3

Authors: We highly appreciate your valuable comments. As per your concerns, all points are addressed carefully and updated the revised manuscript. Thanks again for understanding. 

Dear Reviewer#6

Thank you very much for your encouragement, kind guidance, and support in helping the authors improve the manuscript.

We are truly grateful for your time and support. Kind Regards,

The Authors of the manuscript.

---

## [Editor Report · Decision Letter 6]

26 Dec 2022

PONE-D-20-41034R6Smoke-free status of homes and workplaces among Indian people: Evidence from Global Adult Tobacco Survey Data-2016/2017PLOS ONE

Dear Dr. Alauddin,

Thank you for submitting your manuscript to PLOS ONE. After careful consideration, we feel that it has merit but does not fully meet PLOS ONE’s publication criteria as it currently stands. Therefore, we invite you to submit a revised version of the manuscript that addresses the points raised during the review process.

 Thank you for revising your manuscript based on the comments of the reviewers. I would like to suggest that you language edit your files for onward processing.  

We look forward to receiving your revised manuscript.

Kind regards,

Sandra Boatemaa Kushitor, Ph.D.

Academic Editor

PLOS ONE
---

## [Author Response · Author response to Decision Letter 6]

5 Jan 2023

Authors’ Response to Reviewers

Manuscript ID: PONE-D-20-41034R6

Title: Smoke-free status of homes and workplaces among Indian people: Evidence from Global Adult Tobacco Survey Data-2016/2017

Dear Respected Editor-in-Chief,

We would like to express our gratitude and thanks to the reviewer and the editor for their useful suggestions for improving the manuscript. We have revised the manuscript accordingly. Thank you once again and looking forward to the final acceptance in your esteemed journal.

Journal Requirements:

Authors: Thank you very much for giving us to improve the revised manuscript. We have carefully checked the reference list and observed that all reference was correct. We do not update or remove any paper. We read the whole manuscript very carefully and tried to fix some grammatical errors and marked its highlighted. Thanks again for understanding. 

Dear Editor

Thank you very much for your encouragement, kind guidance, and support in helping the authors improve the manuscript.

We are truly grateful for your time and support. Kind Regards,

The Authors of the manuscript.

---

## [Editor Report · Decision Letter 7]

13 Jan 2023

PONE-D-20-41034R7Smoke-free status of homes and workplaces among Indian people: Evidence from Global Adult Tobacco Survey Data-2016/2017PLOS ONE

Dear Dr. Alauddin,

Thank you for submitting your manuscript to PLOS ONE. After careful consideration, we feel that it has merit but does not fully meet PLOS ONE’s publication criteria as it currently stands. Therefore, we invite you to submit a revised version of the manuscript that addresses the points raised during the review process. Kindly edit the abstract to reflect the change from smoke free policies to smoke free status. I recommend that the manuscript should be reviewed by a professional proofreader.  The authors should attach a letter a of  from the proof reader when submitting the revisions. 

We look forward to receiving your revised manuscript.

Kind regards,

Sandra Boatemaa Kushitor, Ph.D.

Academic Editor

PLOS ONE
---

## [Author Response · Author response to Decision Letter 7]

27 Jan 2023

Manuscript ID: PONE-D-20-41034R7

Title: Smoke-free status of homes and workplaces among Indian people: Evidence from Global Adult Tobacco Survey Data-2016/2017

Dear Respected Editor-in-Chief,

We would like to express our gratitude and thanks to the reviewer and the editor for their useful suggestions for improving the manuscript. We have revised the manuscript accordingly. Thank you once again and looking forward to the final acceptance in your esteemed journal.

Academic Editors:

AE1: Kindly edit the abstract to reflect the change from smoke free policies to smoke free status. 

Authors: Thank you very much for your valuable suggestions and giving us to improve the revised manuscript. As per your suggestions, we have changed the words from “smoke free policies” to “smoke free status” in the abstract of the revised manuscript. Thanks again for understanding. 

AE2: I recommend that the manuscript should be reviewed by a professional proofreader. The authors should attach a letter a of from the proof reader when submitting the revisions. 

Authors: Thank you very much for your valuable concerns. As per your concerns, our revised manuscript has been reviewed by a professor cambridge proofreader and also attached a letter a of from the proof reader during on line submission system. Thanks again for understanding. 

Dear Editor

Thank you very much for your encouragement, kind guidance, and support in helping the authors improve the manuscript.

We are truly grateful for your time and support. Kind Regards,

The Authors of the manuscript.

---

## [Editor Report · Decision Letter 8]

9 Feb 2023

Smoke-free status of homes and workplaces among Indian people: Evidence from Global Adult Tobacco Survey Data-2016/2017

PONE-D-20-41034R8

Dear Dr. Alaudin,

We’re pleased to inform you that your manuscript has been judged scientifically suitable for publication and will be formally accepted for publication once it meets all outstanding technical requirements.

Kind regards,

Sandra Boatemaa Kushitor, Ph.D.

Academic Editor

PLOS ONE
---

## [Editor Report · Acceptance letter]

14 Feb 2023

PONE-D-20-41034R8 

Smoke-free status of homes and workplaces among Indian people: Evidence from Global Adult Tobacco SurveyData-2016/2017 

Dear Dr. Alauddin:

I'm pleased to inform you that your manuscript has been deemed suitable for publication in PLOS ONE. Congratulations! Your manuscript is now with our production department. 

Kind regards, 

on behalf of

Dr. Sandra Boatemaa Kushitor 

Academic Editor

PLOS ONE